# Directing visceral white adipocyte precursors to a thermogenic adipocyte fate improves insulin sensitivity in obese mice

Chelsea Hepler, Mengle Shao, Jonathan Y Xia, Alexandra L Ghaben, Mackenzie J Pearson, Lavanya Vishvanath, Ankit X Sharma, Thomas S Morley, William L Holland, Rana K Gupta*

Touchstone Diabetes Center, Department of Internal Medicine, University of Texas Southwestern Medical Center, Dallas, United States

**Abstract** Visceral adiposity confers significant risk for developing metabolic disease in obesity whereas preferential expansion of subcutaneous white adipose tissue (WAT) appears protective. Unlike subcutaneous WAT, visceral WAT is resistant to adopting a protective thermogenic phenotype characterized by the accumulation of Ucp1+ beige/BRITE adipocytes (termed 'browning'). In this study, we investigated the physiological consequences of browning murine visceral WAT by selective genetic ablation of *Zfp423*, a transcriptional suppressor of the adipocyte thermogenic program. *Zfp423* deletion in fetal visceral adipose precursors (*Zfp423*<sup>loxP/loxP</sup>; *Wt1-Cre*), or adult visceral white adipose precursors (*Pdgfrb*<sup>rtTA</sup>; *TRE-Cre; Zfp423*<sup>loxP/loxP</sup>), results in the accumulation of beige-like thermogenic adipocytes within multiple visceral adipose depots. Thermogenic visceral WAT improves cold tolerance and prevents and reverses insulin resistance in obesity. These data indicate that beneficial visceral WAT browning can be engineered by directing visceral white adipocyte precursors to a thermogenic adipocyte fate, and suggest a novel strategy to combat insulin resistance in obesity.

*For correspondence: Rana. Gupta@UTSouthwestern.edu

**Competing interests:** The authors declare that no competing interests exist.

## Introduction

Adipocytes are critical regulators of energy balance and nutrient homeostasis. White adipocytes serve as the principal site for energy storage in mammals. These cells are characterized by a large unilocular lipid droplet and have the capacity to store or release energy depending on metabolic demand. White adipocytes also produce and secrete numerous cytokines and hormones that impact several aspects of physiology (*Rosen and Spiegelman, 2014*). Eutherian mammals also contain a second major class of adipocytes that function to catabolize stored lipids and produce heat. These thermogenic adipocytes, consisting of brown and beige/BRITE adipocytes, are characterized by their multilocular lipid droplet appearance, high mitochondrial content, and expression of *uncoupling protein 1* (Ucp1) (*Cohen and Spiegelman, 2015*; *Harms and Seale, 2013*). Brown/beige adipocytes have promising therapeutic potential as their activation in the setting of obesity has a profound impact on metabolic health.

White adipose depots are broadly categorized as either subcutaneous or visceral adipose tissue, reflecting their anatomical location. Adipose tissue distribution is a strong predictor of metabolic outcome in obese individuals (*Karpe and Pinnick, 2015*; *Lee et al., 2013*). Visceral adiposity strongly correlates with the development of insulin resistance, diabetes, and cardiovascular disease (*Kissebah et al., 1982*; *Krotkiewski et al., 1983*; *Ohlson et al., 1985*; *Vague, 1956*). Preferential

**eLife digest** Mammals have different types of fat cells in their bodies. White fat cells store energy for later use, and brown and beige fat cells burn energy to help keep the body warm. Individuals who are obese typically have too many white fat cells in and around their belly. This belly fat, also called visceral fat, accumulates around the organs and is believed to contribute to metabolic diseases, such as diabetes and heart disease. Individuals who are obese also have relatively few brown and beige energy-burning fat cells.

Boosting the amount of brown and beige fat in individuals who are obese has been proposed as a potential way to reduce their risk of metabolic disease. One way to do this would be to encourage white visceral fat cells to become more like energy-burning beige or brown fat cells.

Recent research has shown that white fat cells contain higher amounts of a protein called Zfp423 than brown or beige fat cells. This protein turns off the genes that fat cells use to burn energy and so keeps white fat cells in an energy-storing state. Now, Hepler et al. show that genetically modifying mice to turn off the gene that produces Zfp423 specifically in the precursor cells that become white fat cells causes more energy-burning beige cells to appear in their visceral fat.

The genetically modified mice were better able to tolerate cold than normal mice. When placed on a high-fat diet, the modified mice were also less likely to become resistant to the effects of the hormone insulin – a process that can lead to the development of type 2 diabetes and may be linked to heart disease. This suggests that treatments that prevent Zfp423 from working in fat cells could help to treat or prevent diabetes and heart disease in people who are obese. Before such treatments can be developed, further work is needed to investigate how Zfp423 works in more detail, and to confirm that it has the same effects in human fat cells as it does in mice.

expansion of subcutaneous depots is associated with sustained insulin sensitivity (*Manolopoulos et al., 2010*). Engineered rodent models highlight the protective role of subcutaneous adipose tissue. Transgenic animals overexpressing *adiponectin* or *mitoNEET* develop extreme subcutaneous obesity; however, these animals remain metabolically healthy (*Kim et al., 2007*; *Kusminski et al., 2012*).

In humans, the location of visceral adipose tissue itself likely mediates some of its detrimental effects on energy metabolism; lipids, metabolites, and cytokines can drain directly into the portal circulation and affect liver function (*Rytka et al., 2011*). Transplantation studies, cellular studies, and gene expression analyses, suggest that factors intrinsic to these depots may also determine their effect on nutrient homeostasis (*Tran et al., 2008*; *Yamamoto et al., 2010*). Anatomically distinct adipocytes are functionally unique, differing in their ability to undergo lipolysis, lipogenesis, and activate the thermogenic gene program (*Lee et al., 2013*; *Macotela et al., 2012*; *Morgan-Bathke et al., 2015*; *Wu et al., 2012*). Along these lines, lineage analyses reveal that anatomically distinct white adipocytes can originate from developmentally distinct precursor cells, and emerge at different times during development (*Billon and Dani, 2012*; *Chau et al., 2014*). As such, it is now widely believed that visceral and subcutaneous adipocytes represent distinct subtypes of fat cells.

In mice, visceral and subcutaneous white adipose depots differ remarkably in their ability to remodel under physiological conditions (*Hepler and Gupta, 2017*). Upon high-fat diet feeding, visceral adipose depots of mice expand by both adipocyte hypertrophy and through the formation of new adipocytes ('adipogenesis') (*Wang et al., 2013*). Inguinal subcutaneous WAT expands predominantly through adipocyte hypertrophy. The differential capacity for adipogenesis is likely explained by factors present in the local microenvironment (*Jeffery et al., 2016*). Another notable difference between the inguinal and visceral adipose depots in rodents is the capacity to adopt a thermogenic phenotype. Various stimuli, including $\beta$3-adrenergic receptor-agonism and cold exposure, drive the rapid accumulation of beige adipocytes in subcutaneous depots (*Vitali et al., 2012*; *Wu et al., 2012*). Genetic stimulation of subcutaneous beige adipogenesis renders mice resistant to high-fat diet induced obesity and/or diabetes (*Seale et al., 2011*; *Shao et al., 2016*), while inhibition of subcutaneous beiging leads to an earlier onset of insulin resistance during obesity (*Cohen et al., 2014*). Most visceral depots in mice, particularly the gonadal and mesenteric adipose tissues, are relatively

resistant to browning in response to physiological stimuli. With few exceptions (*Kiefer et al., 2012*), most engineered mouse models of white adipose tissue browning exhibit beige cell accumulation preferentially in subcutaneous WAT depots (*Seale et al., 2011*; *Stine et al., 2016*). Visceral WAT depots may harbor mechanisms to suppress thermogenesis to ensure its function as a white, energy-storing, depot.

We previously established a critical role for the transcription factor, *Zfp423*, in the establishment and maintenance of the adipocyte lineage. *Zfp423* is required for fetal differentiation of subcutaneous white adipocytes (*Gupta et al., 2010*; *Shao et al., 2017*). In adult mice, *Zfp423* expression defines a subset of committed mural preadipocytes (*Gupta et al., 2012*; *Vishvanath et al., 2016*). Upon high-fat diet feeding, these mural cells undergo adipogenesis in visceral depots, contributing to adipocyte hyperplasia (*Vishvanath et al., 2016*). *Zfp423* is also expressed in nearly all mature adipocytes; however, its expression is more abundant in white adipocytes than brown adipocytes (*Shao et al., 2016*). In the mature adipocyte, *Zfp423* acts to maintain the energy-storing status of the white adipocyte through suppression of the thermogenic gene program (*Shao et al., 2016*). *Zfp423* likely exerts this function by serving as a transcriptional co-repressor of the brown/beige lineage determining transcription factor, *Ebf2*. The loss of *Zfp423* in mature white adipocytes triggers a robust conversion of white to beige adipocytes in subcutaneous WAT. Amongst the various adipose tissues, visceral WAT expresses the highest levels of *Zfp423*. Importantly, we observed that visceral adipocytes lacking *Zfp423* were also capable of inducing their thermogenic gene program when animals were stimulated pharmacologically with a $\beta3$ adrenergic receptor agonist (*Shao et al., 2016*). This observation affords the possibility of examining whether the thermogenic capacity of visceral white adipose depots can be unlocked under physiological conditions, and whether thermogenic visceral WAT would be ultimately harmful or beneficial to systemic metabolic health.

Here, we describe two mouse models of visceral adipose tissue browning derived through selective ablation of *Zfp423* in visceral adipose precursors. We reveal that fetal visceral white preadipocytes can be redirected to a beige-like adipocyte fate through the loss of *Zfp423*, leading to visceral depot mass reduction and fat redistribution towards subcutaneous depots. The browning of visceral depots improves cold tolerance and protects against the development of insulin resistance and hyperlipidemia in obesity. Moreover, we demonstrate that visceral mural preadipocytes in adult mice can also be directed to a thermogenic cell fate; this leads to beige-like, rather than white, adipocyte hyperplasia, in the expanding visceral WAT depots of diet-induced obese animals. Upon activation by $\beta-3$ adrenergic receptor agonism, these beige-like adipocytes can trigger an amelioration of insulin resistance in obese animals. Together, these data establish *Zfp423* as a physiological suppressor of the thermogenic gene program in visceral WAT and provide proof of concept that the thermogenic capacity of visceral WAT can be induced under physiological conditions through removal of this molecular brake on the thermogenic gene program in adipose precursors. These data also highlight the potential of visceral WAT, much like subcutaneous WAT, to improve nutrient homeostasis in obesity when a thermogenic beige-like phenotype is induced.

## Results

### Selective ablation of *Zfp423* in fetal visceral white adipocyte precursors leads to the formation of thermogenic visceral adipose depots

We hypothesized that the thermogenic capacity of visceral white adipose depots can be unlocked under physiological conditions through removal of *Zfp423*, and that engineered visceral thermogenic adipocytes can drive improvements in systemic metabolic health. To test this, we derived a mouse model in which *Zfp423* is selectively inactivated in visceral, but not subcutaneous, WAT depots or classic brown adipose tissue depots (BAT). These animals were generated by breeding transgenic mice expressing Cre recombinase under the control of the *Wilms Tumor one* locus (*Wt1-Cre*) to animals carrying the floxed *Zfp423* alleles (*Zfp423*^loxP/loxP^) (*Figure 1A*) (*Zfp423*^loxP/loxP^; *Wt1-Cre* animals, herein 'Vis-KO' mice). *Wt1-Cre* mice targets *Wt1*-expressing cells of the embryonic mesothelium, as well as descending cells, which include, adult mesothelial cells and intra-abdominal stromal cells within or surrounding visceral organs. Hastie and colleagues revealed that visceral white adipocytes, but not subcutaneous or classic brown adipocytes, descend from *Wt1*-expressing progenitors and are targeted by *Wt1-Cre* (*Chau et al., 2014*). Specifically, *Wt1-Cre* targets the majority of gonadal

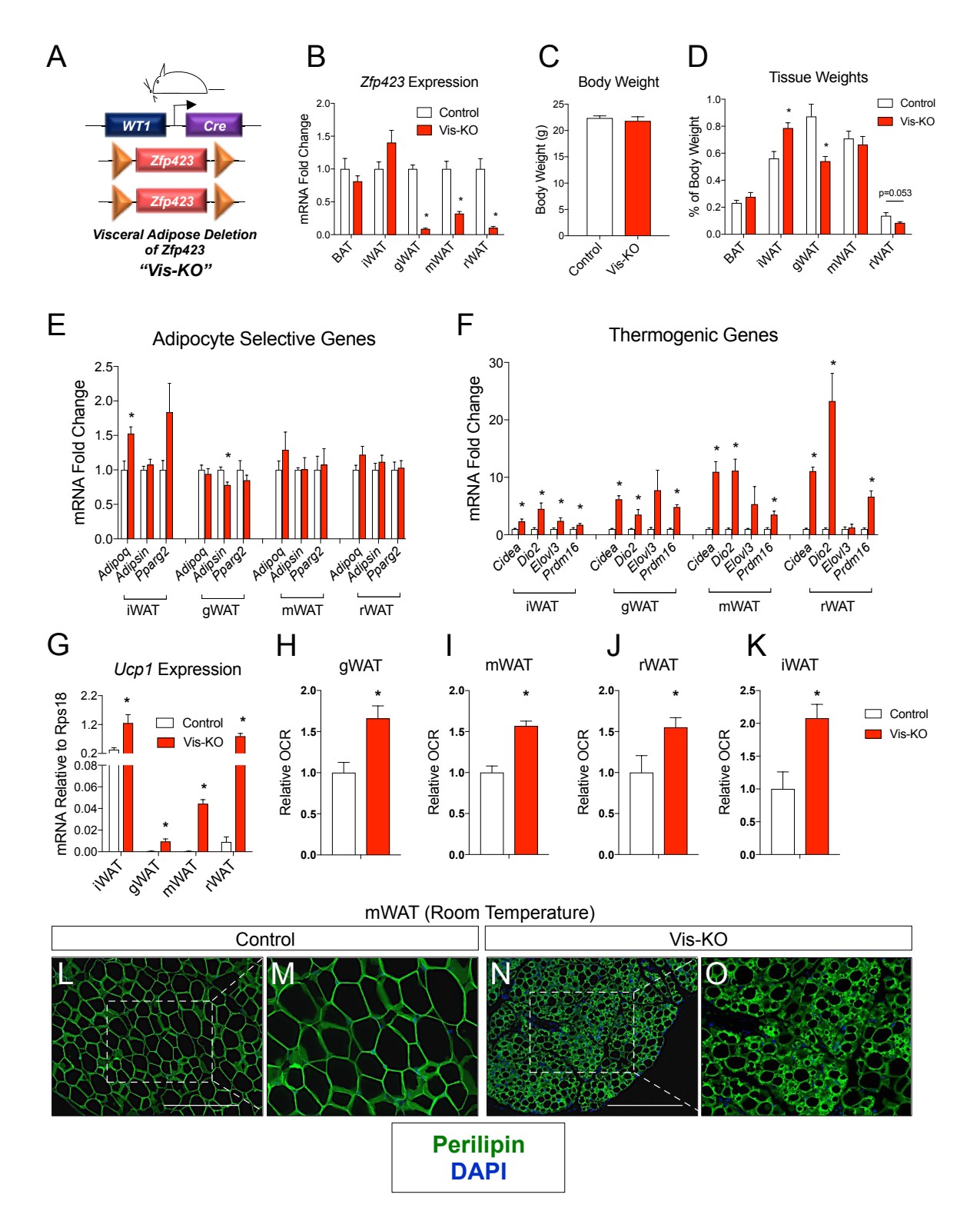

**Figure 1.** Deletion of *Zfp423* in fetal visceral white adipocyte precursors leads to the formation of thermogenic adipocytes in visceral depots. (A) A mouse model of visceral WAT selective ablation of *Zfp423* (Visceral-Knockout or 'Vis-KO') was derived by breeding animals expressing the gene encoding Cre recombinase under the control of the *Wilms Tumor 1* (*WT1*) promoter to animals carrying floxed *Zfp423* alleles (*Zfp423*loxP/loxP). Littermates carrying only *Zfp423*loxp/loxP alleles were used as control animals. (B) Fold change in mRNA levels within brown (BAT), inguinal (iWAT),

*Figure 1 continued on next page*

*Figure 1 continued*

gonadal (gWAT), mesenteric (mWAT), or retroperitoneal (rWAT) adipose tissue of control (white bar) and Vis-KO (red bar) mice at 8 weeks of age. * denotes p<0.05 from unpaired Student's t-test. n = 6 mice. (C) Body weight of control (white bar) and Vis-KO (red bar) mice at 8 weeks of age. n = 6 mice. (D) Fat pad weight (normalized to body weight) of control (white bar) and Vis-KO (red bar) mice at 8 weeks of age. * denotes p<0.05 from unpaired Student's t-test. n = 6 mice. (E– F) Fold change in mRNA levels of mature adipocyte genes (E) and thermogenic genes (F) isolated from whole adipose tissue from control (white bar) and Vis-KO (red bar) mice at 8 weeks of age. * denotes p<0.05 from unpaired Student's t-test. n = 6 mice. (G) mRNA levels (normalized to *Rps18*) of *Ucp1* isolated from whole adipose tissue from control (white bar) and Vis-KO (red bar) mice at 8 weeks of age. * denotes p<0.05 from unpaired Student's t-test. n = 6 mice. (H–K) Relative basal oxygen consumption rates (OCRs) within diced gWAT (H), mWAT (I), rWAT (J), and iWAT (K) from control (white bar) and Vis-KO (red bar) mice at 8 weeks of age. * denotes p<0.05 from unpaired Student's t-test. n = 4 independent replicates pooled from two mice. (L–O) Representative immunofluorescence staining of Perilipin (green) and DAPI (blue) in mWAT sections obtained from control (L, M) and Vis-KO (N, O) mice at 8 weeks of age. Scale bar, 200 μM. Panels M and O represent digital enhancements of the boxed regions shown in L and N, respectively.

The following figure supplement is available for figure 1:

**Figure supplement 1.** (A–L) Representative images of gonadal (gWAT), retroperitoneal (rWAT), and inguinal WAT (iWAT) from control and Vis-KO mice immunostained with antibodies raised against Perilipin (green).

adipocyte precursors and variable numbers of precursors in other visceral depots, including the mesenteric and retroperitoneal depots. Analysis of mRNA expression from multiple fat depots in Vis-KO mice confirmed the deletion of *Zfp423* was selective to visceral WAT depots (*Figure 1B*).

Vis-KO mice were born in the expected Mendelian ratio and appeared grossly indistinguishable from controls. We did not observe a difference in body weight between eight weeks-old Vis-KO mice and littermate controls; however, we did observe a noticeable difference in body fat distribution (*Figure 1C*). In comparison to control animals, Vis-KO mice had smaller gonadal WAT depots and larger subcutaneous inguinal WAT (*Figure 1D*). mRNA levels of adipocyte-selective genes in visceral adipose depots from the Vis-KO were comparable to levels found in control animals (*Figure 1E*); these data suggest that visceral adipocyte differentiation per se does not depend on *Zfp423*. However, in all visceral depots examined, there was a marked increase in the expression of key genes involved in adipocyte thermogenesis, including *Ucp1* (*Figure 1F–G*). This was accompanied by an increase in basal $O_2$ consumption rates in explanted WAT depots (*Figure 1H–J*). Furthermore, some visceral depots (mesenteric and retroperitoneal) lacking *Zfp423* exhibit a multilocular appearance typical of thermogenic adipocytes (*Figure 1L–O*, *Figure 1—figure supplement 1A–L*). Taken together, these results indicate that inactivation of *Zfp423* in fetal visceral white adipocyte precursors leads to the widespread accumulation of beige-like thermogenic adipocytes within visceral adipose depots. Interestingly, despite the fact that *Zfp423* was not inactivated in the subcutaneous inguinal WAT of Vis-KO animals, the thermogenic capacity of this depot was also enhanced (*Figure 1F,G,K*).

## *Zfp423*-deficient visceral adipose precursors differentiate into functional thermogenic adipocytes in vitro

We next asked whether the browning of WAT depots in the Vis-KO animals may occur in a cell-autonomous manner. We isolated the adipose stromal vascular fraction (SVF) from control and Vis-KO mice and performed in vitro adipocyte differentiation assays. SVF cultures obtained from the inguinal WAT of control and Vis-KO animals differentiated to a similar degree, and no significant differences in levels of *Zfp423* or thermogenic genes were observed (*Figure 2A–C*). This is consistent with lack of Cre activity in this depot, and suggests that the increased inguinal WAT beiging in Vis-KO mice in vivo occurs secondary to inactivation of *Zfp423* in the *Wt1* lineage.

Cultures of gonadal WAT SVF from control and mutant animals also underwent adipogenesis to a similar degree under the differentiation conditions utilized (dexamethasone, IMBX, insulin, and rosiglitazone). This was evident by comparable lipid accumulation and mRNA levels of adipocyte-selective genes in differentiated cultures (*Figure 2A,D*). Importantly, *Zfp423* mRNA was almost entirely absent from these cultures, reflecting the activity of the *Wt1-Cre* line in this depot. Cultures of gonadal white adipocytes lacking *Zfp423* robustly activate their thermogenic gene program upon stimulation with forskolin (*Figure 2E*). These changes were accompanied by increased basal,

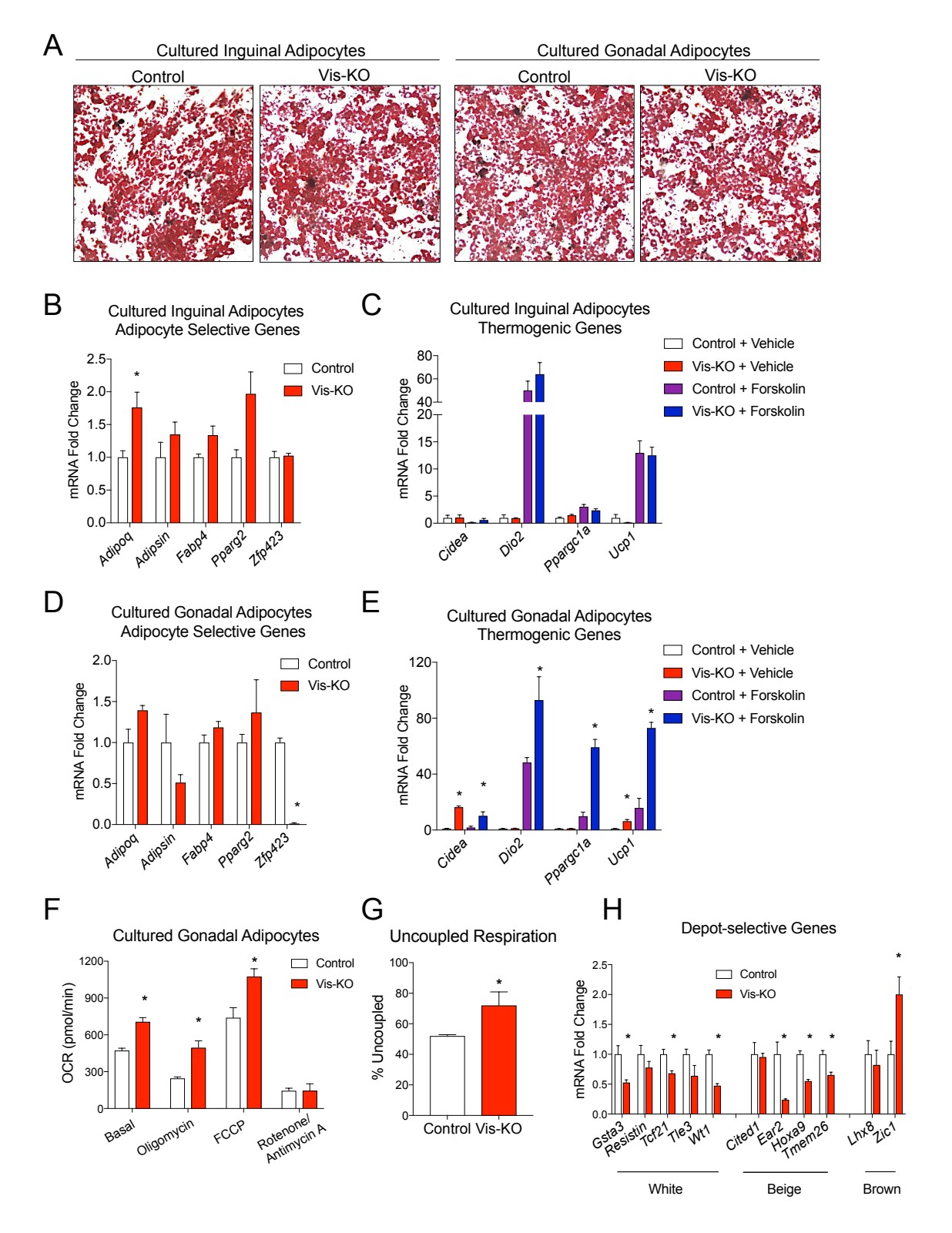

**Figure 2.** *Zfp423*-deficient visceral adipose precursors differentiate into functional thermogenic adipocytes in vitro. (A) Oil Red O staining of adipocytes differentiated in vitro from inguinal and gonadal SVF of control and Vis-KO mice. (B) Fold change in mRNA levels of adipocyte-selective genes in inguinal adipocyte cultures differentiated as shown in A. * denotes p<0.05 from unpaired Student's t-test. n = 4 replicates from two mice. (C) Fold change in mRNA levels of thermogenic genes in inguinal adipocyte cultures differentiated as shown in A and treated with vehicle or forskolin (10 μm)

*Figure 2 continued on next page*

*Figure 2 continued*

for 3 hr. n = 4 replicates from two mice. (D) Fold change in mRNA levels of adipocyte-selective genes in gonadal adipocyte cultures as shown in A. * denotes p<0.05 from unpaired Student's t-test. n = 4 replicates from two mice. (E) Fold change in mRNA levels of thermogenic genes in gonadal adipocyte cultures differentiated as shown in A and treated with vehicle or forskolin (10 μm) for 3 hr. * denotes p<0.05 from unpaired Student's t-test. n = 4 replicates from two mice. (F) Oxygen consumption rates (OCRs) within differentiated cultures of control (white bars) or *Zfp423*-deficient (red bars) gonadal adipocytes in the basal state, and in response to sequential additions of oligomycin (ATP synthase inhibitor), FCCP (chemical uncoupler), and rotenone/antimycin A (complex I and complex III inhibitor). * denotes p<0.05 from unpaired Student's t-test. n = 4 replicates from two mice. (G) Percent uncoupled respiration of differentiated cultures of control (white bar) or *Zfp423*-deficient (red bar) gonadal adipocytes as determined by basal and uncoupled (oligomycin) respiration data from *Figure 2F*. * denotes p<0.05 from unpaired Student's t-test. n = 4 replicates from two mice. (H) Fold change in mRNA levels of white-, beige-, and brown-selective genes in differentiated cultures of control (white bars) or *Zfp423*-deficient (red bars) gonadal adipocytes. * denotes p<0.05 from unpaired Student's t-test. n = 4 replicates from two mice.

uncoupled (oligomycin), and maximal (FCCP) respiration in differentiated gonadal adipocyte cultures from mutant animals (*Figure 2F,G*). These data indicate that *Zfp423*-deficient visceral adipose precursors can differentiate into functional thermogenic adipocytes in a cell-autonomous manner. The cellular phenotype of *Zfp423*-deficient visceral adipocytes is reminiscent of the characterized phenotype of subcutaneous beige adipocyte cultures (*Wu et al., 2012*); therefore, we examined whether *Zfp423*-deficient visceral adipocyte cultures are enriched in the expression of beige-selective transcripts. *Zfp423*-deficient visceral adipocytes expressed lower levels of white adipocyte-selective genes; however, we did not observe a clear pattern of gene expression changes that would indicate a conversion of *Zfp423*-deficient visceral adipocytes to either subcutaneous beige or classic brown fat cells (*Figure 2H*).

## Thermogenic *Zfp423*-deficient visceral adipose tissue is largely distinct from subcutaneous beige adipose tissue

Prior studies of the beige adipocyte determination factor, *Prdm16*, revealed that genetic ablation of *Prdm16* in adipose tissue leads to the loss of subcutaneous beige adipocytes, with inguinal WAT acquiring the molecular properties of visceral fat. Thus, we next asked whether the loss of *Zfp423* reprograms visceral WAT into subcutaneous WAT. Alternatively, visceral WAT lacking *Zfp423* may retain a global visceral phenotype but adopt a thermogenic phenotype reminiscent of classical brown or subcutaneous beige adipose tissues. In order to address this question we obtained and compared global gene expression profiles of adipose depots from control and Vis-KO animals through RNA sequencing (RNA-seq). For this experiment, we treated all animals with a $\beta-3$ adrenergic receptor agonist (CL316,243), or vehicle, for 3 days at thermoneutrality. This treatment allowed us to capture the global gene expression profile of thermogenic adipose depots in their fully active state. CL316,243 treatment led to a much greater increase in mRNA levels of *Ucp1* and most other thermogenic genes examined in visceral WAT depots of Vis-KO mice than in control animals (*Figure 3A,B*). Levels of *Ucp1* mRNA in the visceral depots lacking *Zfp423* nearly reached levels of *Ucp1* found in inguinal WAT following CL316,243 treatment (i.e. subcutaneous beige adipose tissue) (*Figure 3A*). This was accompanied by the widespread appearance of multilocular cells in gonadal WAT with readily detectable Ucp1 protein expression (*Figure 3C–F*). We again assayed the mRNA levels of genes commonly used as white, beige, and classic brown, adipocyte markers. Similar to results obtained from cultured Zfp423-deficient visceral adipocytes, the expression of some white adipocyte-selective genes examined were lower in Vis-KO gWAT; however, we once again did not observe a clear pattern of gene expression changes that would indicate a conversion of *Zfp423*-deficient gWAT to either subcutaneous beige or classic brown adipose tissue (*Figure 3G*).

Principal component analysis and unsupervised hierarchical clustering analysis of RNA-Seq data suggest that the global gene expression profile of *Zfp423*-deficient gWAT is distinct from subcutaneous beige adipose tissue and classic BAT. Instead, the *Zfp423*-deficient visceral WAT more closely resembles gWAT from control animals (*Figure 4A–B*). We next compared the list of transcripts that are differentially expressed between $\beta$3-agonist treated *Zfp423*-deficient gWAT and $\beta$3-agonist treated gWAT of control animals (i.e. genes whose expression reflect engineered thermogenic gWAT, *Figure 4—source data 1*) to the list of transcripts that are differentially regulated between $\beta$-agonist treated iWAT and vehicle-treated iWAT of control animals (i.e. genes whose expression

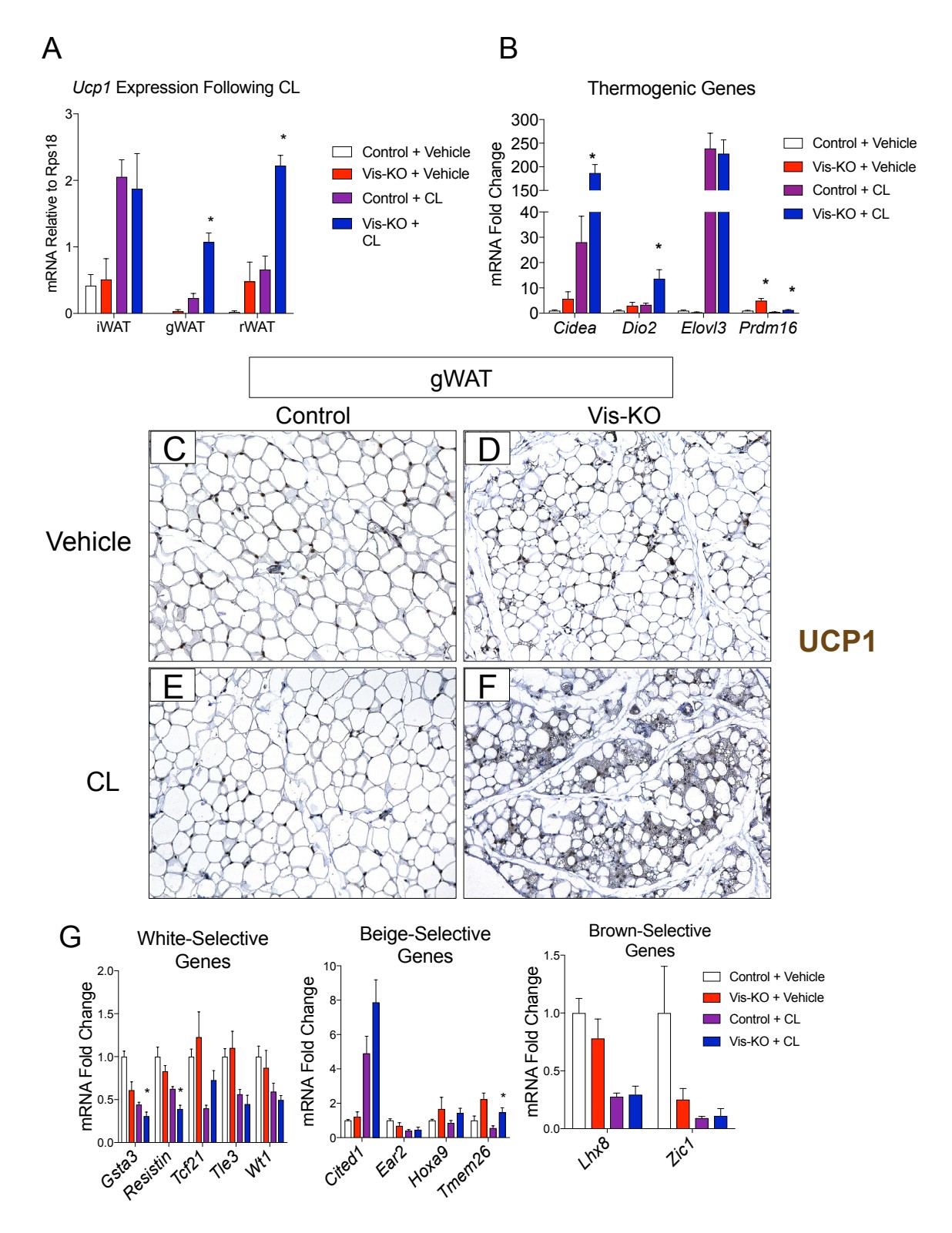

**Figure 3.** Visceral adipose tissue deletion of *Zfp423* leads to enhanced visceral browning upon *β−3* adrenergic receptor agonism at thermoneutrality. (**A**) Control and Vis-KO mice were housed at room temperature (22°C) until 6 weeks of age then transferred to thermoneutral housing conditions (30°C) for 2 weeks. After 2 weeks at thermoneutrality, mice were treated with PBS (vehicle) or CL316,243 (1 mg/kg, intraperitoneal) daily for 3 days. On the 4thday, tissues were harvested for analysis. mRNA levels (relative to *Rps18*) of *Ucp1* in iWAT, gWAT, and rWAT obtained from control and Vis-KO mice

*Figure 3 continued on next page*

*Figure 3 continued*

treated with vehicle or CL316,243 (CL) for 3 days at thermoneutrality. * denotes p<0.05 from unpaired Student's t-test. n = 4–5 mice. (B) Fold change in mRNA levels of thermogenic genes in gWAT obtained from control and Vis-KO mice treated with vehicle or CL316,243 (CL) for 3 days at thermoneutrality. * denotes p<0.05 from unpaired Student's t-test. n = 4–5 mice. (C–F) Representative brightfield images of Ucp1 immunoreactivity (brown staining) in gWAT sections obtained from control or Vis-KO mice treated with vehicle (C,D) or CL316,243 (E,F). (G) Fold change in mRNA levels of white-, beige-, and brown-selective genes in gWAT obtained from control and Vis-KO mice treated with vehicle or CL316,243 (CL) for 3 days at thermoneutrality. * denotes p<0.05 from unpaired Student's t-test. n = 4–5 mice.

reflects beige inguinal adipose tissue, *Figure 4—source data 2*). *Zfp423*-deficiency led to the differential expression of 1735 transcripts in gWAT (FDR $\leq$ 0.05) (*Figure 4C*). Of these 1735 transcripts, 207 genes (12%) overlap with genes differentially regulated following $\beta$3-agonist treatment of iWAT (*Figure 4—source data 3*). Functional classification of these 207 genes by gene ontology analysis indicated that nearly all of these genes are related to mitochondrial function and biogenesis, a hallmark of thermogenic adipocytes (*Figure 4D,E*). These data suggest that *Zfp423*-deficient adipocytes present in Vis-KO animals are visceral adipocytes expressing at least a portion of the thermogenic gene program characteristic of subcutaneous beige adipocytes.

## Thermogenic visceral adipose tissue confers improved cold tolerance and protection from insulin resistance in obesity

Subcutaneous beige adipocytes exert beneficial effects on nutrient homeostasis and can increase energy expenditure. The Vis-KO model affords the opportunity to explore the systemic benefits of unlocking the thermogenic phenotype of visceral adipose depots. To this end, we tested the ability of Vis-KO mice to maintain body temperature in response to cold challenge. Following 1 week of cold exposure, visceral WAT depots of the Vis-KO mice activated their thermogenic gene program (*Figure 5A,B*) and accumulated Ucp1+ multilocular adipocytes (*Figure 5C–J*) to a much greater extent than control animals. We did not observe a significant difference in core body temperature during an acute cold tolerance test of control and Vis-KO animals (*Figure 5K*); however, after four weeks of cold acclimation, the Vis-KO mice maintain higher core body temperature (*Figure 5L*). Collectively, these data demonstrate that inactivation of *Zfp423* is sufficient to unlock the thermogenic activity of visceral WAT depots under postnatal physiological conditions, and that browning of these depots can functionally enhance adaptive thermogenesis.

We also asked whether the thermogenic visceral WAT present in Vis-KO animals confers protection against diet-induced obesity and/or impaired nutrient homeostasis. We administered 8 weeks-old male mice chow or high-fat diet (HFD) (60% calories from fat) for 20 weeks. Over this period, we did not observe a significant difference in body weight between Vis-KO mice and controls (*Figure 6A*). At the time of analysis (8 weeks of HFD feeding), we did not observe a significant difference in adiposity (% body fat), lean mass, or fat distribution (*Figure 6B,C*). Gene expression analysis of dissected tissues revealed that the loss of *Zfp423* expression remained restricted to visceral depots (*Figure 6D*). Levels of *Ucp1* and other genes related to thermogenesis were significantly increased in visceral depots of obese Vis-KO mice (*Figure 6E,F*); however, mRNA levels of adipocyte-selective genes did not appear to be impacted by the loss of *Zfp423* (*Figure 6G*). The later suggests that *Zfp423*-deficiency did not impact de novo visceral adipocyte differentiation that occurs over this period of HFD feeding. The increase in thermogenic gene expression appeared functionally significant; rates of basal and maximal oxygen consumption were significantly elevated in the visceral depots of obese Vis-KO mice when compared to controls (*Figure 6H–J*). We again observed a slight increase in the expression of thermogenic genes in the inguinal WAT of mutant animals; however, the metabolic activity of the inguinal WAT depot did not significantly differ from inguinal WAT of obese control mice (*Figure 6K*).

We also examined whether the observed increase in visceral WAT oxygen consumption impacts whole-body levels of energy expenditure. We assessed energy expenditure and food intake in control and knockout animals fed HFD for 8 weeks. At this point, body weights and body composition were comparable between the two groups of animals (*Figure 7A,B*); however, we observed a modest, but statistically significant, increase in $O_2$ consumption, heat production, and $CO_2$ production, in the *Zfp423*-deficient mice (*Figure 7C–F*). We observed a slight increase in food intake over the dark

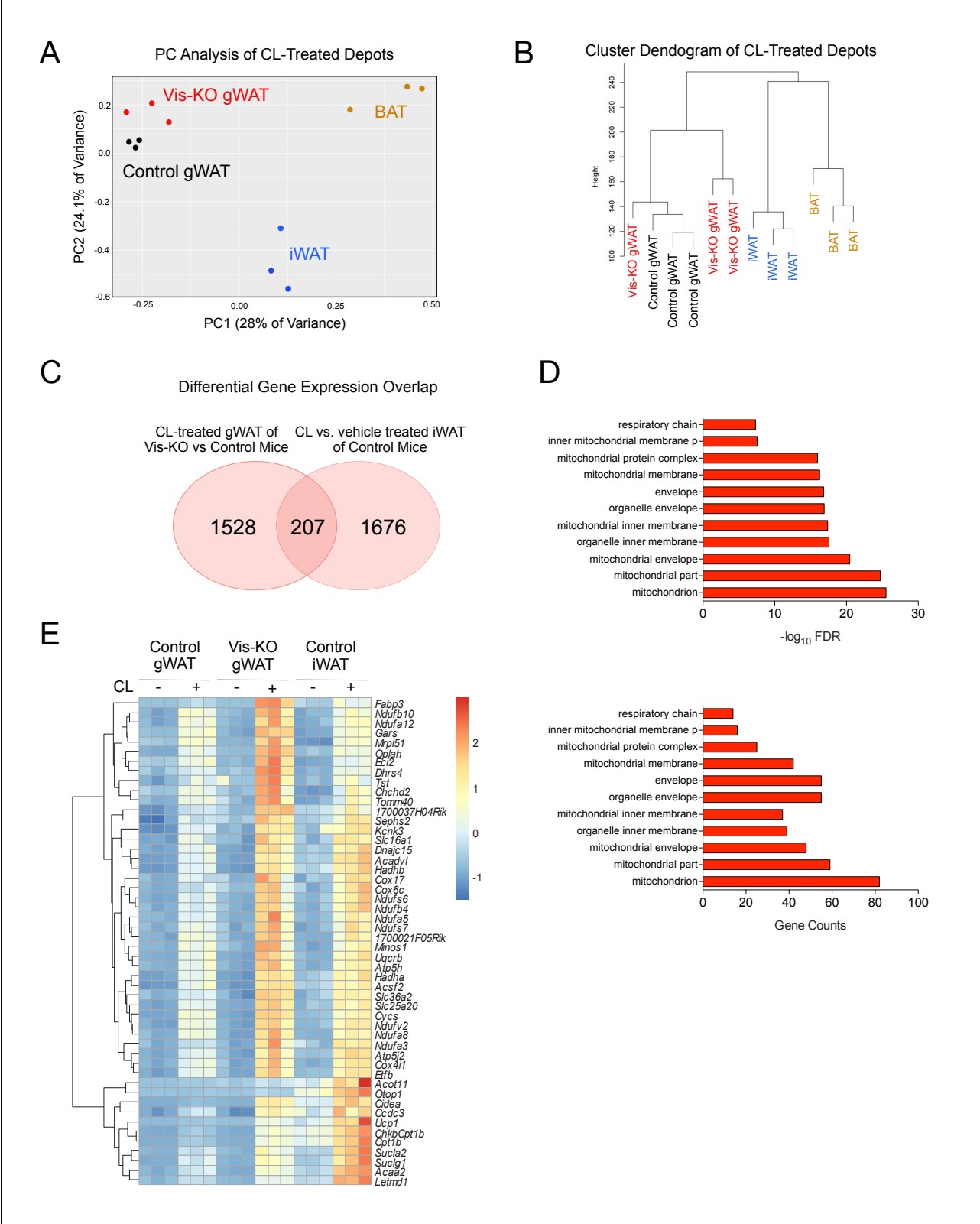

**Figure 4.** Thermogenic *Zfp423*-deficient visceral adipose tissue is largely distinct from subcutaneous beige adipose tissue. (**A**) RNA-sequencing was performed on mRNA libraries derived from gonadal WAT (gWAT) of vehicle- and β3 agonist (CL)-treated control and Vis-KO mice, as well as from inguinal WAT and interscapular BAT of vehicle- and CL-treated control mice. Principle component (PC) analysis of sequencing data obtained from CL-treated animals. n = 3 sequenced libraries for each condition. (**B**) Unsupervised clustering dendrogram of adipose depot samples. (**C**) Venn diagram

*Figure 4 continued on next page*

*Figure 4 continued*

depicting overlap in 1) transcripts that are differentially expressed between *β*3-agonist treated *Zfp423*-deficient gWAT and *β*3-agonist treated gWAT of control animals, and 2) transcripts that are differentially regulated between *β*-agonist treated iWAT and vehicle-treated iWAT of wild-type animals. See *Figure 4—source data 1*, *Figure 4—source data 2*, *Figure 4—source data 3*. (**D**) Gene ontology analysis (GOTERM_CC_FAT) of the 207 overlapping genes shown in D. (**E**) Heatmap of top 50 induced genes of the 207 highlighted in C.
The following source data is available for figure 4:

**Source data 1.** Genes whose expression is significantly altered in gonadal WAT of Vis-KO CL316,243 treated mice.
**Source data 2.** Genes whose expression is significantly altered in Inguinal WAT of CL316,243 treated mice.
**Source data 3.** List of 207 genes highlighted in *Figure 4C*.

cycle; however, these differences did not reach statistical significance (*Figure 7G*). Moreover, overall locomotor activity measurements and respiratory exchange ratios were similar between control and knockout animals (*Figure 7H,I*). These new data indicate that visceral WAT deletion of *Zfp423*, and subsequent visceral browning, enhances energy expenditure, albeit not a strong enough degree to drive a robust difference in body weight.

Further metabolic phenotyping of control and mutant animals revealed that the beige-like phenotype of visceral WAT in diet-induced obese Vis-KO mice was associated with striking improvements in nutrient homeostasis. Obese Vis-KO mice exhibited significantly better glucose tolerance test than obese control animals; excursions in blood glucose and serum insulin following glucose challenge were substantially lower in Vis-KO animals (*Figure 8A,B*). Insulin tolerance tests suggest greater systemic insulin sensitivity in the Vis-KO mice (*Figure 8C*). We further explored the impact of visceral WAT browning on systemic glucose metabolism by performing hypersulinemic-euglycemic clamp assays. The glucose infusion rate needed to maintain euglycemia (~138 mg/dl) was increased in Vis-KO mice (*Figure 8D*). This demonstrates an increase in whole-body insulin sensitivity, consistent with the aforementioned insulin tolerance tests. Importantly, endogenous glucose output was suppressed much more efficiently during the basal and clamped states in Vis-KO mice, likely reflecting improved insulin sensitivity at the level of the liver (*Figure 8E*). These data are supported by liver gene expression analysis from a separate cohort of animals. The mRNA levels of key gluconeogenic genes in the Vis-KO livers are significantly lower than corresponding levels in livers from control animals after 8 weeks of HFD feeding (*Figure 8F*). Furthermore, visceral adipose-selective browning phenotype is associated with lower serum triglyceride levels (*Figure 8G*). $^{14}$C-2-deoxyglucose tracing revealed that the rate of whole-body glucose disposal did not significantly differ between control and Vis-KO mice (*Figure 8H*); however, glucose uptake was enhanced in the visceral rWAT depot, correlating with where the largest degree of visceral beiging is observed in this model (*Figure 8I*). All together, these data demonstrate visceral deletion of *Zfp423* leads to enhanced insulin sensitivity and reduced hepatic glucose output. Moreover, these data suggest that engineered thermogenic visceral adipose tissue, much like subcutaneous beige adipose tissue, can defend against the development of impaired glucose and lipid homeostasis in obesity.

## Inactivation of *Zfp423* in adult mural cells leads to beige, rather than white, adipocyte hyperplasia in diet-induced obesity

We previously demonstrated that adipocytes emerging in expanding visceral, but not subcutaneous, WAT depots of diet-induced obese mice arise through de novo adipocyte differentiation from mural progenitors expressing *Pdgfrb* (*Hepler et al., 2017*; *Vishvanath et al., 2016*). Whether these preadipocytes can be redirected to a thermogenic adipocyte fate, thereby driving beige-like adipocyte hyperplasia rather than white adipocyte hyperplasia, has been unclear. To address this, we generated a model that allows for doxycycline-inducible inactivation of *Zfp423* in *Pdgfrb*-expressing mural cells (inducible mural cell *Zfp423* knockout, or 'iMural-KO') (*Figure 9A*). The iMural-KO model was achieved by breeding *Pdgfrb*^rtTA^ transgenic mice to animals expressing Cre recombinase under the control of a tetracycline responsive element (*TRE-Cre*) and carrying floxed *Zfp423* alleles (*Zfp423*^loxP/loxP^). We also bred the Cre-dependent Rosa26R *loxP*-mtdTomato-*loxP*-mGFP (mT/mG) fluorescent

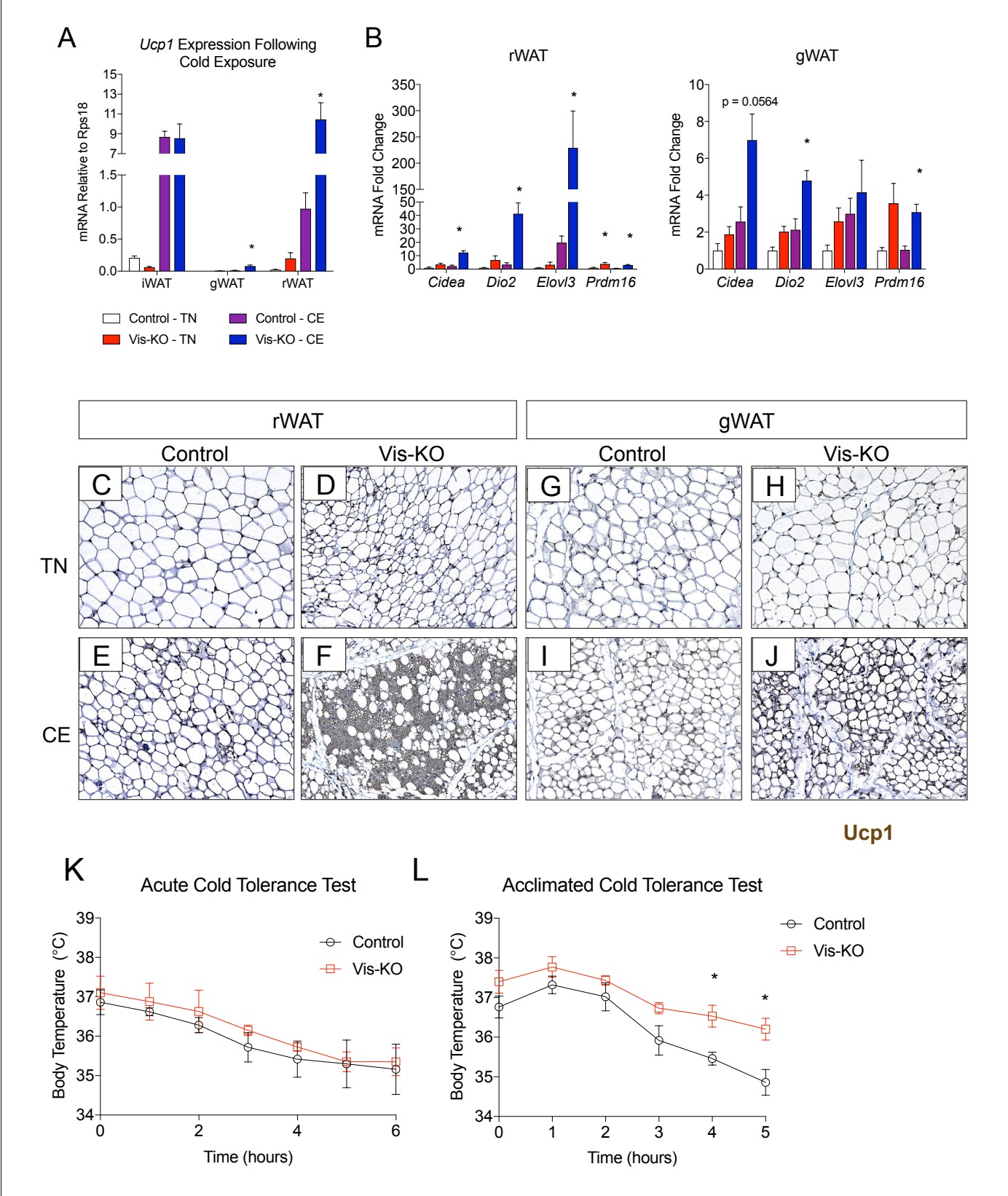

**Figure 5.** *Zfp423*-deficiency enables cold-induced visceral WAT browning. (**A**) Control and Vis-KO mice were housed at room temperature (22°C) until 6 weeks of age then transferred to thermoneutral housing (30°C) for 2 weeks. After 2 weeks at thermoneutrality, animals were transferred to 6°C or left at
*Figure 5 continued on next page*

*Figure 5 continued*

thermoneutrality for seven full days. On the 8th day, tissues were harvested for analysis. mRNA levels (relative to *Rps18*) of *Ucp1* in iWAT, gWAT, and rWAT from control and Vis-KO mice cold exposed (CE) or housed at thermoneutrality (TN) for 7 days. * denotes p<0.05 from unpaired Student's t-test. n = 4–5 mice. (B) Fold change in mRNA levels of thermogenic genes in rWAT and gWAT from control and Vis-KO mice cold exposed (CE) or housed at thermoneutrality (TN) for 7 days. * denotes p<0.05 from unpaired Student's t-test. n = 4–5 mice. (C–J) Brightfield images of Ucp1 immunostaining (brown) within rWAT (C–F) and gWAT (G–J) sections obtained from control or Vis-KO mice following cold exposure (CE) or housing at thermoneutrality (TN). (K) Body temperature at indicated time points following transfer of mice from thermoneutrality to cold (6℃). n = 4–5 mice. (L) Body temperature at indicated time points during a cold tolerance test after 4 weeks of acclimation to cold (6℃). * denotes p<0.05 from unpaired Student's t-test. n = 4–5 mice.

reporter allele into the model; this allowed for the fate-mapping of targeted mural cells. Treatment of adult animals with doxycycline leads to Cre dependent inactivation of mural cell *Zfp423* (*Figure 9B*), and permanent fluorescent-tagging (mGFP) of Pdgfr$\beta^+$ cells. Importantly, mature adipocytes present at the time of doxycycline treatment are not targeted (*Figure 9B*). Only those adipocytes that descend from mural cells following HFD feeding are *Zfp423*-deficient and will express mGFP.

Following doxycycline treatment, we fed control and iMural-KO mice a HFD for 8 weeks (*Figure 9C*). After 8 weeks of HFD feeding, the body weights of mutant animals were indistinguishable from controls (*Figure 9D*). De novo white adipocyte differentiation occurred in gWAT of both control and mutant animals; newly derived adipocytes were indicated by the expression of mGFP (*Figure 9E–J*). However, we did not observe a statistically significant difference in the number of mGFP+ adipocytes between control and iMural-KO mice (*Figure 9K*). Overall, ~5–15% of gonadal adipocytes present in the gWAT of these obese animals descended from mural cells. Despite the fact that *Zfp423* expression identifies mural adipose progenitors, these data suggest that adult mural progenitors do not require *Zfp423* in vivo for their ability to undergo adipocyte differentiation in response to HFD feeding.

*Zfp423*-deficient adipocytes originating from mural progenitors in the obese iMural-KO model were readily identified by mGFP expression; however, these cells were not multilocular (*Figure 9H,I, J*). Moreover, the glucose tolerance of obese iMural-KO mice was similar to that of control animals (*Figure 9L*). Our previous study revealed that the thermogenic activity of *Zfp423*-deficient adipocytes was dependent on active $\beta$-adrenergic signaling (*Shao et al., 2016*); it is well appreciated that rodent obesity is associated with augmented $\beta$-adrenergic receptor signaling (*Collins et al., 1999*; *Collins and Surwit, 2001*). Therefore, we reasoned that the *Zfp423*-deficient adipocytes present in these in obese mice would require a stimulus to fully activate their thermogenic function in this setting. To test this, we utilized osmotic pumps to deliver the $\beta$3-adrenergic receptor agonist, CL316,243, daily at a dose of 1 mg/kg/day for four continuous weeks. The agonist was given to obese control and iMural-KO mice after 8 weeks of HFD feeding (*Figure 10A*). During the 4 weeks of the $\beta$3-agonist treatment, both control and iMural-KO animals exhibited a similarly mild decrease in body weight (*Figure 10B*). However, after four weeks of $\beta$3-agonist treatment, the gonadal adipose depot mass of iMural-KO mice was significantly smaller than the gWAT mass of treated control mice (*Figure 10C*). Immunohistochemistry for mGFP expression revealed that $\beta$3-agonist treatment did not induce the formation of any additional gonadal or inguinal adipocytes from the *Pdgfrb* lineage (*Figure 10D–H*, *Figure 10—figure supplement 1A,B*). We again did not observe a difference in the numbers of mGFP-labelled adipocytes between control and knockout animals (*Figure 10—figure supplement 1A,B*). Mural-cell derived adipocytes in the $\beta$3-agonist treated control animals remained unilocular; however, most mGFP+ adipocytes present in $\beta$3-agonist treated knockout mice were now multilocular (*Figure 10D–H*). On average, ~6% of gonadal adipocytes appeared multilocular in the iMural-KO animals; very few, if any, gonadal multilocular adipocytes were present in $\beta$3-agonist treated control mice (*Figure 10—figure supplement 1C*). Levels of *Ucp1* mRNA were strongly elevated in visceral depots of the $\beta$3-agonist treated iMural-KO mice (*Figure 10—figure supplement 1E*). On the other hand, the percentage of inguinal adipocytes appearing multilocular appeared low (~3%) and comparable between control and knockout animals (*Figure 10—figure supplement 1D*). However, despite the low and comparable levels of mural cell adipogenesis, the iWAT from $\beta$3-agonist treated knockout animals had relatively higher levels of *Ucp1* mRNA in comparison

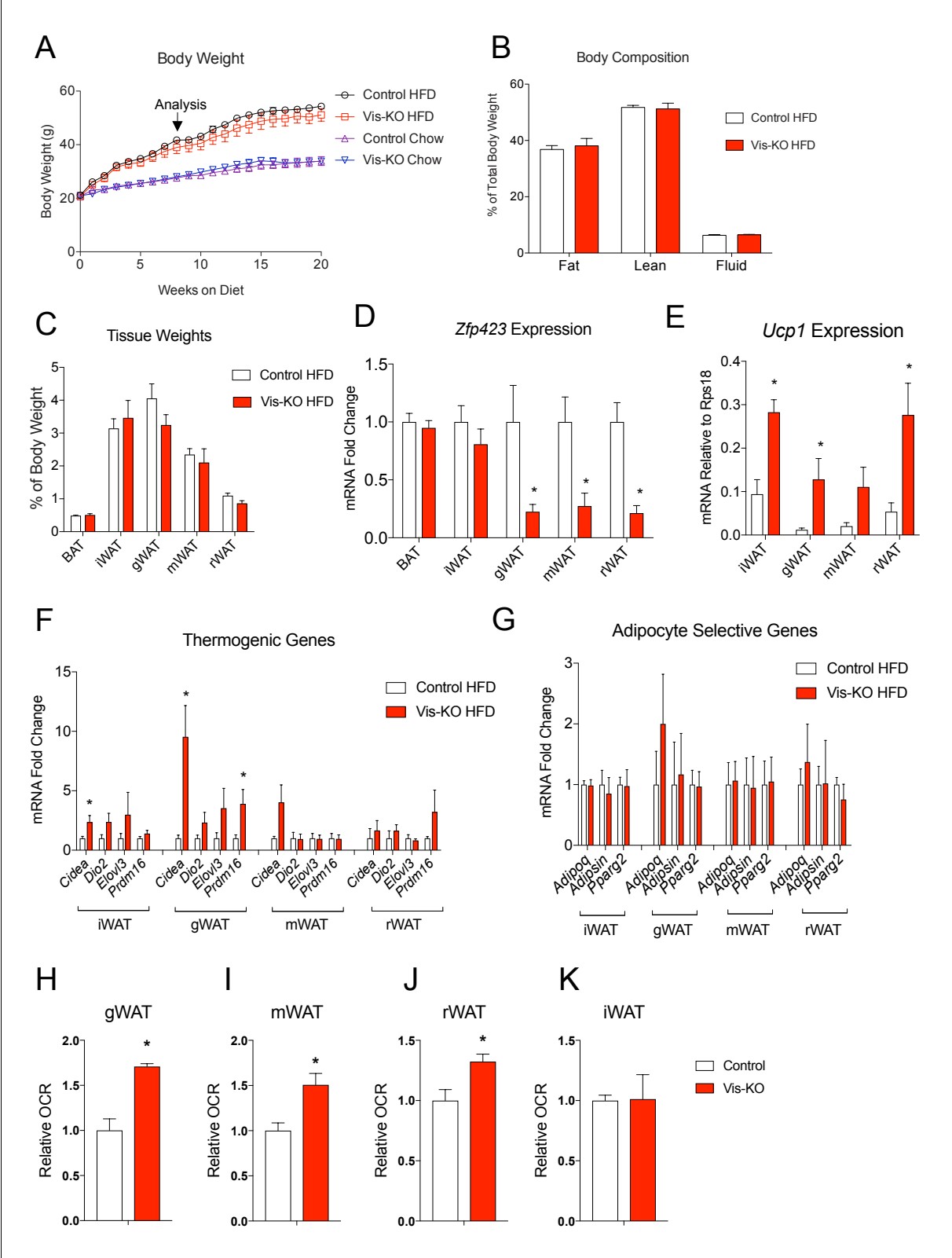

**Figure 6.** Obese mice lacking *Zfp423* in visceral adipose tissue display enhanced visceral adipose tissue thermogenesis. (**A**) Weekly measurements of body weights of 8 weeks-old control and Vis-KO mice fed a standard chow diet (Chow) or high fat diet (HFD) for 20 weeks. n = 6–7 mice per group. (**B**) Total fat mass, lean mass, and water mass (normalized to body weight) of control (white bars) and Vis-KO (red bars) mice after 8 weeks of HFD feeding. n = 6–7 mice. (**C**) Fat depot mass (normalized to body weight) of control (white bars) and Vis-KO (red bars) mice after 8 weeks of HFD feeding. n = 6–7

*Figure 6 continued on next page*

*Figure 6 continued*

mice. (D) Relative mRNA levels of *Zfp423* in BAT, iWAT, gWAT, mWAT, and rWAT isolated from control (white bars) and Vis-KO (red bars) mice after 8 weeks of HFD feeding. * denotes $p<0.05$ from unpaired Student's t-test. n = 6–7 mice per group. (E) mRNA levels (normalized to *Rps18*) of *Ucp1* within adipose depots of control (white bars) and Vis-KO (red bars) mice after 8 weeks of chow or HFD feeding. * denotes $p<0.05$ from unpaired Student's t-test. n = 6–7 mice. (F–G) Relative mRNA levels of thermogenic genes (F) and adipocyte-selective genes (G) in iWAT, gWAT, mWAT, and rWAT isolated from control (white bars) and Vis-KO (red bars) mice after 8 weeks of HFD feeding. * denotes $p<0.05$ from unpaired Student's t-test. n = 6–7 mice. (H–K) Relative basal oxygen consumption rates (OCRs) within gWAT (H), mWAT (I), rWAT (J), and iWAT (K) isolated from control (white bars) and Vis-KO (red bars) mice after 8 weeks of chow or HFD feeding. * denotes $p<0.05$ from unpaired Student's t-test. n = 4 independent replicates pooled from two mice.

to controls (*Figure 10—figure supplement 1E*). The recruitment of this relatively low number of beige-like adipocytes appeared sufficient to drive an increase in basal respiration of adipose tissues (*Figure 10—figure supplement 1F*). This active thermogenic phenotype of the visceral WAT correlated with markedly improved glucose tolerance and insulin sensitivity observed in $\beta$3-agonist iMural-KO mice (*Figure 10I–K*). All together, these data indicate that visceral mural preadipocytes in adult mice can be directed to a dormant beige-like phenotype in the expanding visceral WAT depots of diet-induced obese animals. Upon activation by $\beta$3 adrenergic receptor agonism, these beige-like adipocytes appear to drive an improvement in insulin sensitivity in obese animals.

## Discussion

It is now certain that adult humans have appreciable amounts of thermogenic adipose tissue consisting of brown and beige adipocytes (*Cypess et al., 2013*; *Shinoda et al., 2015*; *van Marken Lichtenbelt et al., 2009*; *Virtanen et al., 2009*). Upon activation, thermogenic adipose tissue in lean adults can impact glucose and lipid homeostasis (*Chondronikola et al., 2014*, *2016*; *Cypess et al., 2015*); however, it still remains unclear as to whether sufficient amounts of thermogenic adipose tissue are present in obese individuals to exert beneficial therapeutic effects, even when fully activated. As such, there is tremendous interest in identifying strategies to increase the mass of functional thermogenic adipose tissue in obese patients with metabolic syndrome. To this end, a number of studies have now identified pathways/factors that can drive the natural formation of subcutaneous beige adipocytes or classic brown adipocytes (*Harms and Seale, 2013*). The negative impacts of visceral expansion during obesity on metabolic health make visceral fat a prime target for therapeutic intervention; however, effective strategies to improve the health of visceral WAT health have yet to emerge. In particular, whether browning of visceral depots, much like the browning of subcutaneous adipose tissue, would exert beneficial metabolic effects has been unclear.

We recently reported that *Zfp423* is a suppressor of the thermogenic gene program in adipocytes. Using this discovery as a tool, we reveal here that the thermogenic potential of visceral adipose depots in mice can be unlocked through removal of *Zfp423*. Importantly, these data provide proof of concept that white adipose precursors in adult animals can be redirected to a beige-like adipocyte fate and improve insulin sensitivity in obesity. Overall, we observed beneficial effects of visceral WAT browning on nutrient homeostasis and cold tolerance. Nevertheless, we cannot exclude the possibility that inducing the thermogenic capacity of these depots may have detrimental effects under other physiological conditions not examined. Elevated expression of *Zfp423* in visceral white adipocytes provides one explanation as to how visceral WAT depots resist adopting a thermogenic phenotype; however, it remains unclear as to why visceral WAT depots would adopt these anti-thermogenic mechanisms.

Inactivation of *Zfp423* in visceral WAT gives rise to thermogenic adipocytes that share properties of subcutaneous beige adipocytes and classic brown adipocytes. In particular, *Zfp423* deletion in visceral WAT unlocks a functionally significant portion of the thermogenic gene program characteristic of activated ($\beta$3 adrenergic receptor activated) beige adipose tissue. Nevertheless, global gene expression profiling suggests that *Zfp423*-deficient visceral WAT largely retains a visceral WAT molecular signature, rather than adopting the global molecular program characteristic of subcutaneous WAT. Additional gene expression studies of isolated visceral and inguinal Ucp1+ adipocytes will be needed to fully characterize the similarities and differences between the anatomically distinct

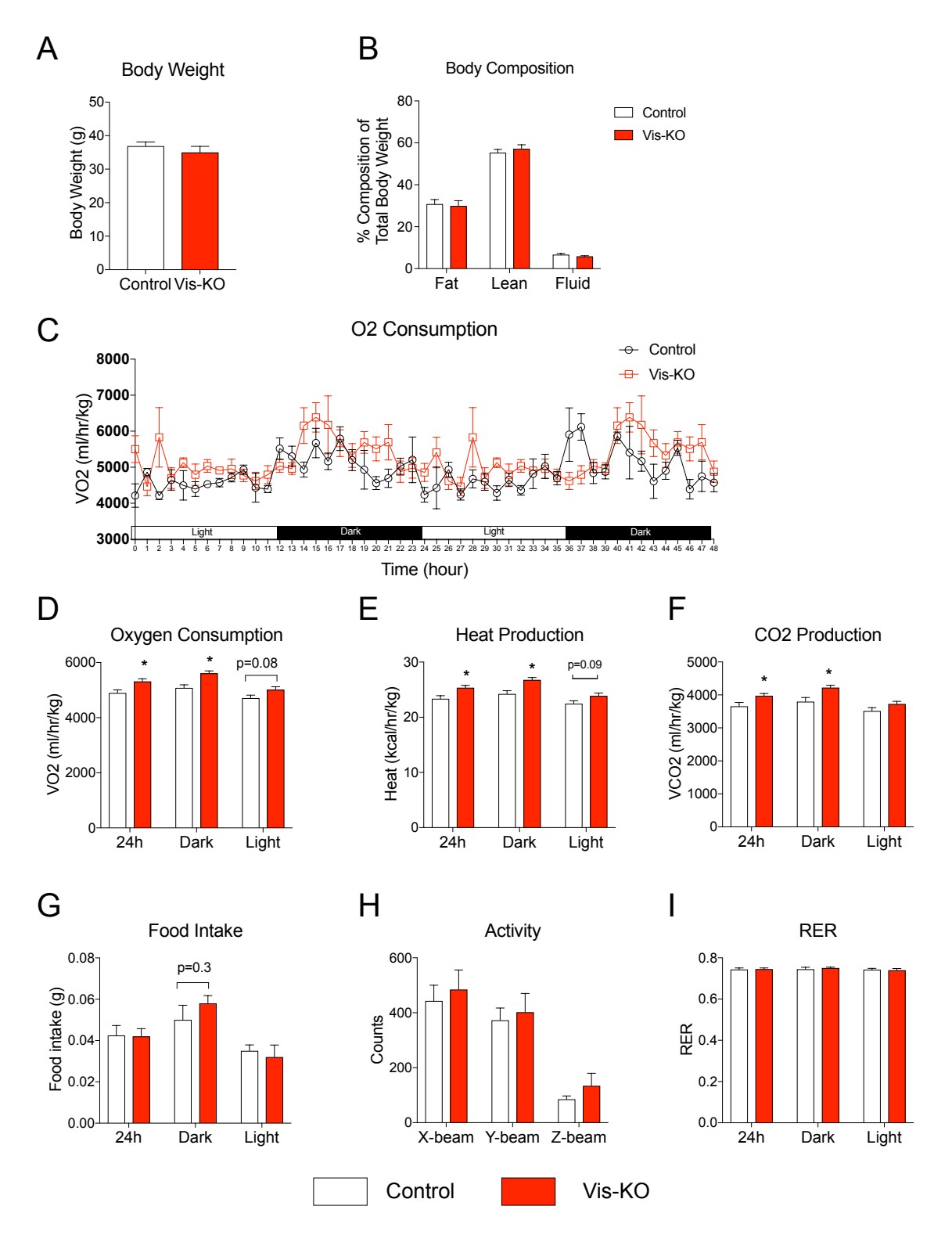

**Figure 7.** Browning of visceral adipose tissues is associated with increased energy expenditure. (A) Body weight of control (white bar) and Vis-KO (red bar) mice after 8 weeks of high fat diet feeding. n = 4–5 mice. (B) Total fat mass, lean mass, and water mass (normalized to body weight) of control (white bars) and Vis-KO (red bars) mice after 8 weeks of HFD feeding. n = 4–5 mice. (C) $O_2$ consumption during two complete 12 hr light-dark cycles of control and Vis-KO mice following 8 weeks of high fat diet feeding. n = 4–5 mice. (D–G) Average $O_2$ consumption (D), heat production (E), $CO_2$

*Figure 7 continued on next page*

*Figure 7 continued*

production (**F**), and food intake (**G**) during the 24 hr, day, and night cycle in control (white bars) and Vis-KO (red bars) mice following 8 weeks of high fat diet. Bars represent averages from over the course of the 5 day measurement. * denotes p<0.05 from unpaired Student's t-test. n = 4–5 mice. (**H**) Average 24 hr X-beam, Y-beam, and Z-beam breaks of control (white bars) and Vis-KO (red bars) mice following 8 weeks of high fat diet. Bars represent averages from over the course of the 5 day measurement. n = 4–5 mice. (**I**) Average RER during the 24 hr, day, and night cycle in control (white bars) and Vis-KO (red bars) mice following 8 weeks of high fat diet. Bars represent averages from over the course of the 5 day measurement. n = 4–5 mice.

thermogenic fat cells. During preparation of this manuscript, Kirichok and colleagues reported the existence of two distinct types of thermogenic beige adipocytes present in visceral depots of mice stimulated with the $\beta$3-adrenergic receptor agonist for 10 days (*Bertholet et al., 2017*). In particular, Bertholet et al. revealed that most thermogenic adipocytes in visceral WAT are devoid of Ucp1 protein and instead employ futile creatine cycling for thermogenesis. *Zfp423*-deficient visceral adipocytes express Ucp1 protein; however, the precise contribution of Ucp1-mediated uncoupling vs. creatine cycling in these cells remains unclear.

Our data here highlight the ability of thermogenic adipocytes, even within the visceral compartment, to drive vast improvements in nutrient homeostasis in obese mice, without impacting body weight. Moreover, the mural cell knockout model of *Zfp423* even suggests that a relatively low frequency of activated multilocular *Zfp423*-deficient visceral adipocytes (5–10% of adipocytes in $\beta$3-agonist treated obese iMural-KO mice) can lead to improved insulin sensitivity. This is surprising given the opinion that such small numbers of beige cells are not likely to confer significant benefit (*Kalinovich et al., 2017*). We, of course, cannot rule out the possibility that *Zfp423* deficiency may impact the visceral adipose phenotype in a multitude of ways that influence energy metabolism. It is notable that in the Vis-KO model, the data largely point to the liver as a major site of improved insulin sensitivity. Previously reported work on *Prdm16* and subcutaneous WAT beiging suggest a strong connection between subcutaneous beige cells and liver health (*Cohen et al., 2014*). For the field at large, it remains an open question as to how brown/beige adipocytes exert beneficial effects on systemic metabolism, independent of their impact on body weight. This unique model may serve as a tool to explore the mechanisms of how visceral WAT browning leads to improved glucose metabolism.

We consistently observed a modest, yet statistically significant, induction of the thermogenic gene program in the inguinal WAT depots of both genetic models. There is a limit to our ability to interpret these results from the iMural-KO mice. Inguinal adipocyte differentiation is not impacted in this model; however, *Zfp423* is still inactivated in mural cells of this depot. It is not possible to exclude additional roles for *Zfp423* in mural cells of iWAT or other tissues. Nevertheless, the dependency of the improved insulin sensitivity in obese iMural-KO mice on $\beta$3-adrenergic receptor activation does suggest that the increased numbers of thermogenic visceral adipocytes, at least in part, contribute to the phenotypes observed here. In the Vis-KO model, inactivation of *Zfp423* is limited to visceral WAT depots. The subcutaneous WAT phenotype in this model is thus not cell autonomous and appears secondary to the visceral phenotype. It is possible that *Zfp423*-deficient visceral adipocytes produce circulating adipokines that influence systemic metabolism and thermogenesis. In fact, more and more 'Batokines' have recently emerged (*Lynes et al., 2017*; *Svensson et al., 2016*; *Thomou et al., 2017*; *Villarroya et al., 2017*). It is also possible that secondary browning of subcutaneous WAT may be triggered through central effects mediated by a visceral WAT to brain neural relay.

A multitude of studies now support the idea that visceral adipocytes are developmentally, molecularly, and functionally, distinct from subcutaneous adipocytes. Our prior work revealed a critical role for *Zfp423* in 3T3-L1 adipogenesis and in the regulation of *Pparg* and the fetal formation of subcutaneous white adipocytes in vivo (*Gupta et al., 2010*; *Shao et al., 2017*). Our work here reveals that *Zfp423* is dispensable for the differentiation of visceral adipocytes within those visceral depots examined. Thus, *Zfp423* expression defines mural white preadipocytes within visceral depots of adult mice; however, other factors can compensate for its absence in the initial stages of adipocyte differentiation, but not in suppressing the thermogenic gene program. The lack of impact on adipocyte differentiation was unexpected; however, this result is perhaps not surprising in light of recent studies of another pro-adipogenic transcription factor, *Cebpa*. In vitro, *Cebpa* is amongst the most

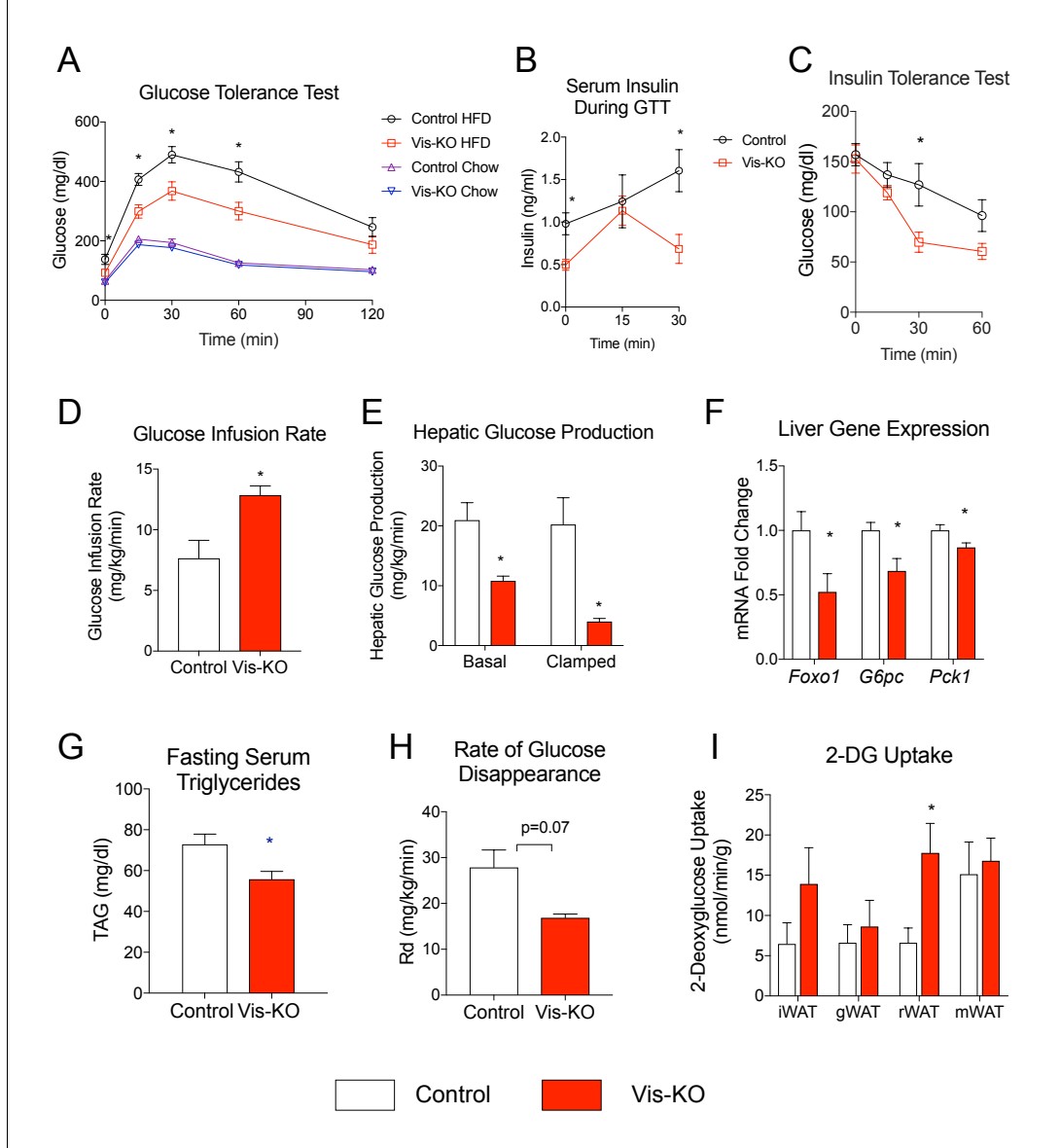

**Figure 8.** Mice lacking visceral adipose *Zfp423* are protected against insulin resistance in obesity. (**A**) Intraperitoneal glucose tolerance tests of control and Vis-KO mice after 8 weeks of chow or HFD feeding. * denotes p<0.05 from unpaired Student's t-test. n = 6–7 mice. (**B**) Serum insulin levels of control and Vis-KO mice after 8 weeks of HFD feeding during the glucose tolerance test shown in A. * denotes p<0.05 from unpaired Student's t-test. n = 6–7 mice. (**C**) Insulin tolerance tests of control and Vis-KO mice after 8 weeks of HFD feeding. * denotes p<0.05 from unpaired Student's t-test. n = 6–7 mice. (**D–E**) Glucose infusion rate (**D**) and basal and clamped hepatic glucose production (**E**) during hyperinsulinemic-euglycemic clamp experiments performed on conscious, unrestrained control (white bars) and Vis-KO (red bars) mice after 8 weeks of HFD feeding. * denotes p<0.05 from unpaired Student's t-test. n = 5–6 mice. (**F**) Relative mRNA levels of gluconeogenic genes in the livers of control (white bars) and Vis-KO (red bars) mice after 8 weeks of HFD feeding. * denotes p<0.05 from unpaired Student's t-test. n = 6–7 mice. (**G**) Serum triglyceride levels in control (white bars) and Vis-KO (red bars) mice after 8 weeks of HFD feeding. * denotes p<0.05 from unpaired Student's t-test. n = 6–7 mice. (**H**) Glucose disappearance rate during hyperinsulinemic-euglycemic clamp experiments performed on conscious, unrestrained control (white bars) and Vis-KO (red bars) mice after 8 weeks of HFD feeding. n = 5–6 mice. (**I**) 2-deoxyglucose uptake quantification of iWAT, gWAT, rWAT, and mWAT during hyperinsulinemic-euglycemic clamp experiments performed on conscious, unrestrained control (white bars) and Vis-KO (red bars) mice after 8 weeks of HFD feeding. * denotes p<0.05 from unpaired Student's t-test. n = 5–6 mice.

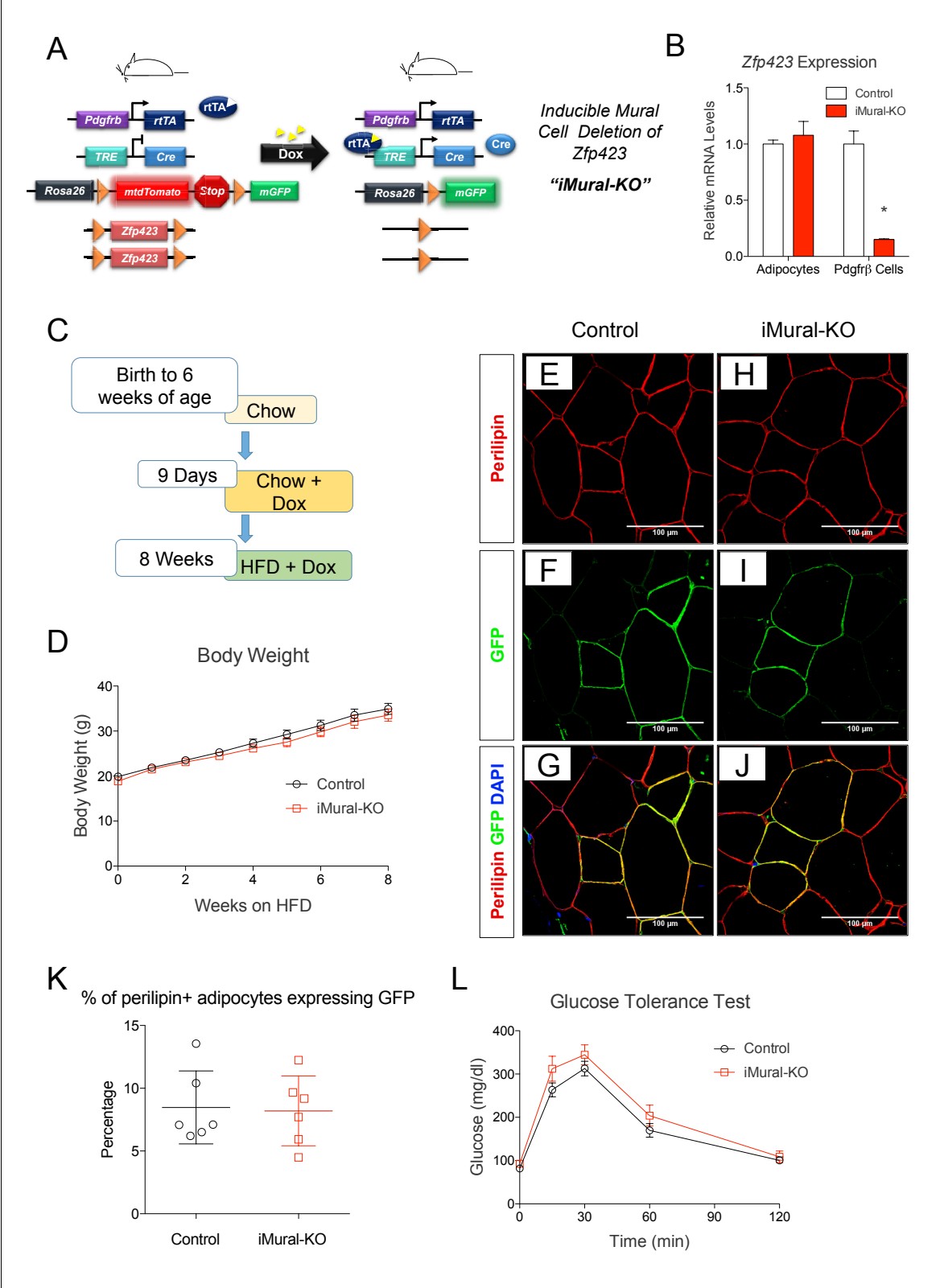

**Figure 9.** Adult mural cell deletion of *Zfp423* does not impact visceral white adipogenesis associated with high fat diet feeding. (**A**) Doxycycline (Dox) inducible deletion of *Zfp423* in adult mural cells (inducible-Mural-Knockout or 'iMural-KO') is achieved by breeding *Pdgfrb*^rtTA transgenic mice to animals expressing Cre recombinase under the control of a tetracycline response element (*TRE-Cre*), the *Rosa26R loxP*-mtdTomato-*loxP*-mGFP (mT/mG) fluorescent reporter allele, and the floxed *Zfp423* alleles (Zfp423^loxP/loxP). Animals carrying only *Pdgfrb*^rtTA, *TRE-Cre*, and Rosa26R^mT/mG alleles were
*Figure 9 continued on next page*

Figure 9 continued

used as controls. (B) Validation of the iMural-KO model. Relative mRNA levels of *Zfp423* in purified adipocytes and purified Pdgfr*β*⁺ cells from WAT obtained from control (white bars) and iMural-KO (red bars) mice at 8 weeks of age. * denotes p<0.05 from unpaired Student's t-test. n = 4–5 mice. (C) Mice were maintained at room temperature and fed standard chow diet until 6 weeks of age. Animals were then switched to Dox-containing chow diet for 9 days in order to induce *Zfp423* deletion and mGFP expression in mural cells. Animals were then maintained on a Dox-containing high-fat diet (HFD) for 8 weeks then sacrificed for analysis. (D) Weekly body weight measurements of control and iMural-KO mice during HFD feeding. n = 6–8 mice. (E–J) Confocal images of immunostained gWAT obtained from control (E–G) and iMural-KO (H–J) mice fed high fat diet for 8 weeks. (K) Quantification of mGFP-expressing adipocytes (mGFP⁺; perlipin⁺) observed in randomly chosen 10X magnification fields of gWAT sections obtained from control (black circles) and iMural-KO (red squares) mice following 8 weeks of HFD feeding. n = 6 mice. (L) Glucose tolerance tests of control and iMural-KO mice after 8 weeks of HFD feeding. n = 6–8 mice.

critical adipogenesis factors; however, in vivo, it appears essential for postnatal, but not fetal, terminal differentiation of fat cells (*Wang et al., 2015*). Thus, despite the expression of *Cebpa* throughout the adipose lineage, there appears to be temporal requirements for this particular factor. These data, along with other studies, highlight the emerging concept that distinct transcriptional regulatory mechanisms govern fetal vs. adult adipogenesis (*Jeffery et al., 2015*; *Wang et al., 2015*). Our data here further highlight the complexity in the regulation of adipogenesis in vivo by demonstrating the depot-specific requirements of pro-adipogenic transcription factors.

*Zfp423* expression in the adipose lineage appears highly regulated. Levels of *Zfp423* in white adipocytes are reduced following cold exposure and *β*-adrenergic receptor agonism, and increased in brown adipose depots undergoing a 'whitening' transformation with age or in obesity (*Shao et al., 2016*). The dynamic expression of *Zfp423*, along with the loss of function data from our genetic mouse models, reveals *Zfp423* as a critical physiological suppressor of the adipocyte thermogenic gene program. Future studies into the regulation of *Zfp423* expression in adipocytes and/or mural adipose precursors may lead to novel strategies to unlock the thermogenic potential of visceral adipose tissue and combat the chronic metabolic defects associated with visceral obesity.

## Materials and methods

### Animals

All animal experiments were performed according to procedures approved by the UTSW Animal Care and Use Committee. *Pdgfrb*^rtTA transgenic mice (C57BL/6-Tg(Pdgfrb-rtTA)58Gpta/J; JAX 028570; RRID:IMSR_JAX:028570) were previously described (*Vishvanath et al., 2016*). *TRE-Cre* (B6. Cg-Tg(tetO-cre)1Jaw/J; JAX 006234; RRID:IMSR_JAX:006234), *Rosa26R*^mT/mG (B6.129(Cg)-*Gt(ROSA) 26Sor*^tm4(ACTB-tdTomato,-EGFP)Luo/J; JAX 007676; RRID:IMSR_JAX:007676), and WT1-Cre (*Wt1*^tm1(EGFP/ cre)Wtp/J; JAX 010911; RRID:IMSR_JAX:010911) mice were obtained from Jackson Laboratories. *Zfp423*^loxP/loxP mice were a gift from Dr. S. Warming (Genentech) (*Warming et al., 2006*). Mice were maintained on a 12-hr light/dark cycle in a temperature-controlled environment (room temperature, 22°C; thermoneutrality, 30°C; cold exposure, 6°C). Mice were given free access to food and water, and maintained on a standard chow diet, Dox-containing chow (600 mg/kg doxycycline, Bio-Serv, S4107), or a dox-containing HFD (600 mg/kg dox, 60% kcal% fat, BioServ, S5867). For acute *β*−3 adrenergic agonist administration, mice were transferred to 30°C chambers for two weeks then injected intraperitoneally with vehicle or CL316243 (1 mg/kg/day) for 3 days. For chronic *β*−3 adrenergic agonist administration, mice were anesthetized by 2% isoflurane, and Alzet osmotic minipumps filled with vehicle (PBS) or CL 316243 (1mg/kg/24 hr) were implanted subcutaneously in the dorsal region of the animals.

### Histological analysis

Adipose tissues were harvested from perfused (4% paraformaldehyde) adult mice. Paraffin processing and embedding was performed by the Molecular Pathology Core Facility at UTSW. Indirect immunofluorescence was performed as previously described (*Vishvanath et al., 2016*). Antibodies used for immunofluorescence include: anti-GFP 1:700 (Abcam ab13970, RRID:AB_300798), anti-perilipin 1:1500 (Fitzgerald 20R-PP004, RRID:AB_1288416), anti-chicken Alexa 488 1:200 (Invitrogen, RRID:AB_142924), anti-guinea pig Alexa 647 1:200 (Invitrogen, RRID:AB_141882), and anti-guinea

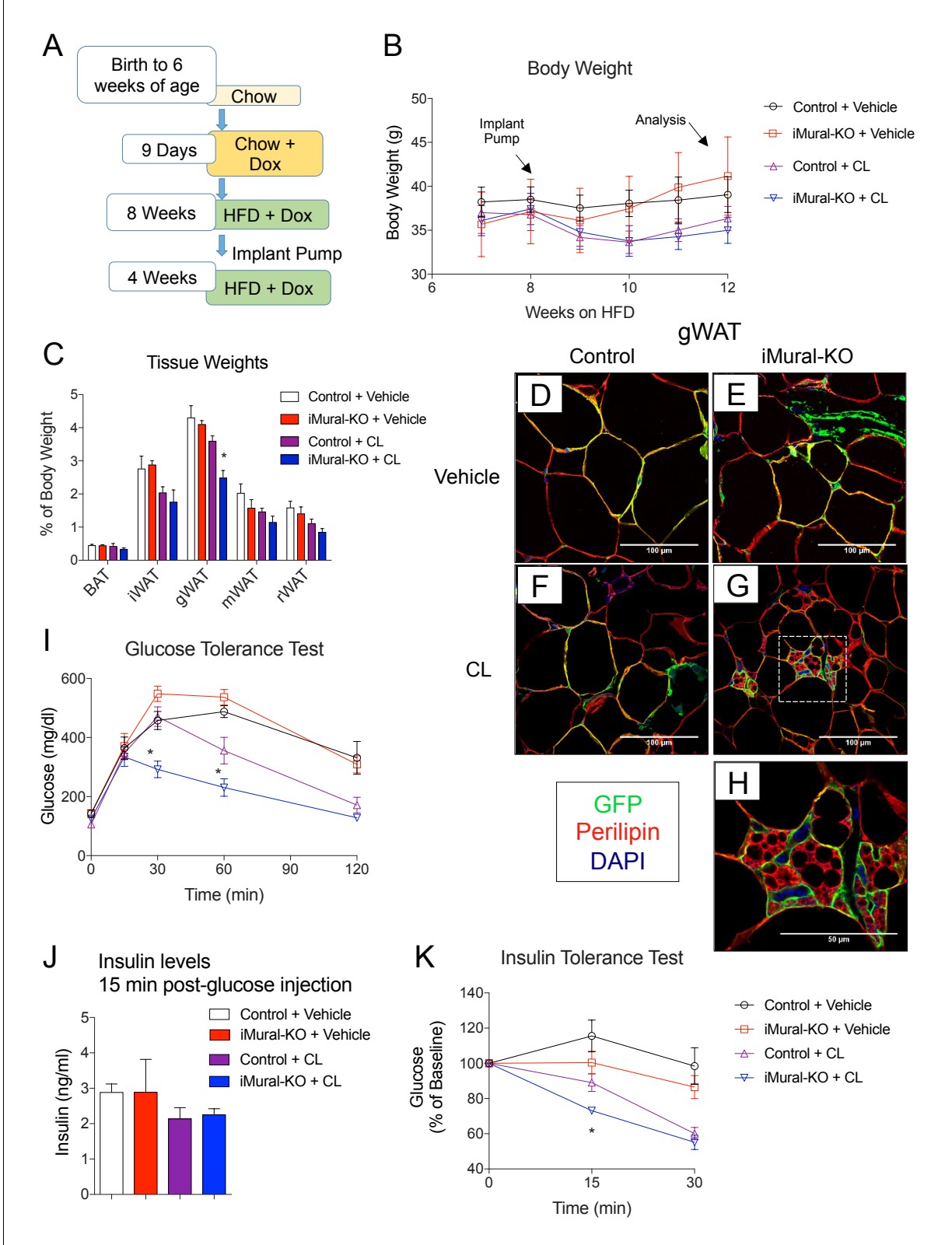

**Figure 10.** Inactivation of *Zfp423* in adult mural cells leads to beige, rather than white, adipocyte hyperplasia in diet-induced obesity. (**A**) Chow-fed 6 weeks-old animals were first administered Dox-containing chow for 9 days in order to inactive *Zfp423* and induce permanent mGFP expression in *Pdgfrb*+ cells. Mice were then switched to a high-fat diet (HFD) containing Dox. After 8 weeks of Dox-HFD, the mice were treated daily with CL316,243 (1 mg/kg/24 hr) or vehicle (PBS) for 4 weeks. (**B**) Weekly body weight measurements of obese control and iMural-KO mice following vehicle or

*Figure 10 continued on next page*

*Figure 10 continued*

CL316,243 administration. n = 6–8 mice. (C) Fat depot weights (normalized to body weight) of control and iMural-KO mice after vehicle or CL316,243 administration. * denotes p<0.05 from unpaired Student's t-test. n = 6–8 mice. (D–G) Representative confocal images of Perilipin (red), mGFP (green) and DAPI (blue) immunostaining of gWAT sections obtained from control and iMural-KO mice administered vehicle (D, E), or CL316,243 (F, G). (H) Digital enhancement of region outlined in (G). (I) Glucose tolerance tests of control and iMural-KO mice following vehicle or CL316,243 administration. * denotes p<0.05 from unpaired Student's t-test. n = 6–8 mice. (J) Serum insulin levels measured during intraperitoneal glucose tolerance tests of control and iMural-KO mice following vehicle or CL316,243 administration. n = 6–8 mice. (K) Insulin tolerance tests of control and iMural-KO mice following vehicle or CL316,243 administration. * denotes p<0.05 from unpaired Student's t-test. n = 6–8 mice.

The following figure supplement is available for figure 10:

**Figure supplement 1.** Quantification of beige adipocyte hyperplasia in iMural-KO mice.

pig Alexa 488 1:200 (Invitrogen, RRID:AB_142018). For Ucp1 immunohistochemistry, paraffin-embedded sections were incubated with anti-Ucp1 (Abcam ab10983; 1:500, RRID:AB_2241462), followed by secondary and tertiary signal amplification and detection using biotinylated anti-rabbit secondary (Vector BA-1100, RRID:AB_2336201), HRP-conjugated streptavidin (Dako), and DAB substrate (Thermo). For quantification of adipocyte hyperplasia, paraffin sections were stained with perilipin and GFP by indirect immunofluorescence. The number of GFP+ perilipin+ and GFP- perilipin+ adipocytes were counted on 8–10 randomly selected 10X images of stained WAT depots. A total of 3,000–5,000 perilipin+ adipocytes were counted from each mouse. Each data point represents the percentage of GFP+ perilipin+ adipocytes from one mouse.

## Isolation of adipose tissue SVF and FACS

SVF was isolated as previously described (*Shao et al., 2016*). Briefly, minced adipose tissue was placed in digestion buffer (100 mM HEPES pH 7.4, 120 mM NaCl, 50 mM KCl, 5 mM glucose, 1 m $CaCl_2$, 1.5% BSA, and 1 mg/mL collagenase D (Roche 11088882001) and incubated in a 37°C shaking water bath for 2 hr. The mixture was then passed sequentially through a 100 µm cell strainer then a 40 µm cell strainer. Cells were blocked in 2% FBS/PBS containing anti-mouse CD16/CD32 Fc Block (clone 2.4G2; 1:200; RRID:AB_394657), then incubated with primary antibodies (anti-CD31 clone 390 1:200, RRID:AB_312903; anti-CD45 clone 30-F11 1:200, RRID:AB_312971; anti-CD140b clone APB5 3:200, RRID:AB_2268091). Cells were sorted using a FACSAria™ flow cytometer (UTSW Flow Cytometry Core Facility).

## Adipocyte differentiation assays

Adipose tissue SVF was isolated as described above. Cells were plated onto collagen-coated dishes and incubated at 10% $CO_2$. Gonadal SVF was maintained in growth media (60% pH7–7.4 low glucose DMEM, 40% pH 7.25 MCDB201 (Sigma M6770)), supplemented with 2% FBS (Fisher Scientific 03-600-511 Lot FB-002) 1% ITS premix (Insulin-Transferrin-Selenium) (BD Bioscience 354352), 0.1 mM L-ascorbic acid-2-2phosphate (Sigma A8960-5G), 10 ng/mL FGF basic (R&D systems 3139-FB-025/CF), Pen/Strep, and gentamicin. Inguinal SVF was maintained in DMEM/F12 (Invitrogen) supplemented with Glutamax, 10% FBS, Pen/Strep, and gentamicin. Upon reaching confluence, cultures were incubated with the adipogenesis induction cocktail (growth media supplemented with 5 mg/ml insulin, 1 µM dexamethasone, 0.5 mM isobutylmethylxanthine, and 1 µM rosiglitazone) for 48 hr. After 48 hr, the cells were maintained in growth media supplemented with 5 mg/ml insulin and 1 µM rosiglitazone until harvest.

## Oil red O staining

Differentiated cells were fixed in 4% PFA for 15 min at room temperature then washed twice with water. Cells were incubated in Oil Red O working solution (2 g Oil red O in 60% isopropanol) for 10 min to stain accumulated lipids. Cells were then washed three times with water before bright field images were acquired.

## Gene expression analysis

Relative mRNA levels were determined by quantitative PCR using SYBR Green chemistry. Values were normalized to Rps18 levels using the $\Delta\Delta$-Ct method. Unpaired Student's t-test was used to evaluate statistical significance. All primer sequences are listed in *Table 1*. mRNA library preparation and RNA-sequencing was performed by the McDermott Center Sequencing Core at UT Southwestern. Total RNA used for library preparation was extracted from gonadal, inguinal, and brown adipose tissues of Control and Vis-KO mice after 3 days of treatment with CL-316,243 at thermoneutrality, as described above. Sequencing was performed on an Illumina HiSeq 2500 and reads were mapped to the mouse genome (mm10). Analysis was performed by the UT Southwestern McDermott Bioinformatics Core using Cufflinks/Cuffdiff software. Genes with an FDR < 0.05 were considered significantly differentially expressed between groups compared. Heatmaps were generated using the Pheatmap package in RStudio (v3.3). The cluster dendogram was generated using Hierarchical Cluster Analysis in RStudio (v3.3). Gene Ontology analysis was performed using the DAVID Functional Annotation tool (https://david.ncifcrf.gov/) on differentially expressed genes between groups compared. Functional annotation for gene ontology using the GOTERM_CC_FAT category was selected and biological processes were assessed for statistical significance. All raw sequencing data has been deposited to Gene Expression Omnibus (https://www.ncbi.nlm.nih.gov/geo/query/acc.cgi?acc=GSE98132).

**Table 1.** qPCR primer sequences utilized for gene expression analysis

| Gene | Forward 5′−3′ | Reverse 5′−3′ |
| --- | --- | --- |
| Adipoq | AGATGGCACTCCTGGAGAGAA | TTCTCCAGGCTCTCCTTTCCT |
| Adipsin | CTACATGGCTTCCGTGCAAGT | AGTCGTCATCCGTCACTCCAT |
| Cidea | TCCTATGCTGCACAGATGACG | TGCTCTTCTGTATCGCCCAGT |
| Cited1 | AACCTTGGAGTGAAGGATCGC | GTAGGAGAGCCTATTGGAGATGT |
| Dio2 | CATTGATGAGGCTCACCCTTC | GGTTCCGGTGCTTCTTAACCT |
| Ear2 | CCACAAAGCAGACAGGGAAAC | GCATGAGGCAAGCATTAGGAC |
| Elovl3 | GTGTGCTTTGCCATCTACACG | CTCCCAGTTCAACAACCTTGC |
| Fabp4 | GATGAAATCACCGCAGACGAC | ATTCCACCACCAGCTTGTCAC |
| Foxo1 | GGCACTCCAAAACAGGACTTG | AAGAAATGGCAGAGGGAGGAG |
| G6pc | GGGCTGTTTGAGGAAAGTGTG | TATCCGACAGGAGGCTGGTAA |
| Gsta3 | AGATCGACGGGATGAAACTGG | CAGATCCGCCACTCCTTCT |
| Hoxa9 | CCCCGACTTCAGTCCTTGC | GATGCACGTAGGGGTGGTG |
| Lhx8 | GAGCTCGGACCAGCTTCA | TTGTTGTCCTGAGCGAACTG |
| Pck1 | TGTCTGTCCCATTGTCCACAG | AAGGTAAGGAAGGGCGGTGTA |
| Pgc1α | GCACCAGAAAACAGCTCCAAG | CGTCAAACACAGCTTGACAGC |
| Pparg2 | GCATGGTGCCTTCGCTGA | TGGCATCTCTGTGTCAACCATG |
| Prdm16 | ACACGCCAGTTCTCCAACCTGT | TGCTTGTTGAGGGAGGAGGTA |
| Resistin | AAGAACCTTTCATTTCCCCTCCT | GTCCAGCAATTTAAGCCAATGTT |
| Rps18 | CATGCAAACCCACGACAGTA | CCTCACGCAGCTTGTTGTCTA |
| Tcf21 | CCCTGAAAGTGGACTCCAACA | GCTGAGCGGGCTTTTCTTAGT |
| Tle3 | GAGACTGAACACAATCCTAGCC | GGAGTCCACGTACCCCGAT |
| Tmem26 | AGGGGCTTCCTTAGGGTTTTC | CCGTCTTGGATGAAGAAGCTG |
| Ucp1 | TCTCAGCCGGCTTAATGACTG | GGCTTGCATTCTGACCTTCAC |
| Wt1 | ATAGGCCAGGGCATGTGTATG | CTGGTGCCTTGCTCTCTGATT |
| Zfp423 | CAGGCCCACAAGAAGAACAAG | GTATCCTCGCAGTAGTCGCACA |
| Zic1 | CTGTTGTGGGAGACACGATG | CCTCTTCTCAGGGCTCACAG |

**Table 2.** Statistical information.

| Figure | N (sample size) | Statistical test method | Description | | p-value |
|---|---|---|---|---|---|
| 1B | Control n = 6 | Unpaired Student's t test | gWAT | | 4.56E-08 |
| | Vis-KO n = 6 | | mWAT | | 0.0019 |
| | | | rWAT | | 0.0002 |
| 1D | Control n = 6 | Unpaired Student's t test | iWAT | | 0.0043 |
| | Vis-KO n = 6 | | gWAT | | 0.0051 |
| 1E | Control n = 6 | Unpaired Student's t test | iWAT | Adipoq | 0.0111 |
| | Vis-KO n = 6 | | gWAT | Adipsin | 0.004 |
| 1F | Control n = 6 | Unpaired Student's t test | iWAT | Cidea | 0.012678087 |
| | Vis-KO n = 6 | | | Dio2 | 0.015258263 |
| | | | | Elovl3 | 0.035976522 |
| | | | | Prdm16 | 0.043549713 |
| | | | gWAT | Cidea | 6.91848E-06 |
| | | | | Dio2 | 0.017046004 |
| | | | | Prdm16 | 9.5932E-05 |
| | | | mWAT | Cidea | 0.008754783 |
| | | | | Dio2 | 0.013198702 |
| | | | | Prdm16 | 0.018964447 |
| | | | rWAT | Cidea | 4.6109E-07 |
| | | | | Dio2 | 0.000934576 |
| | | | | Prdm16 | 0.000260661 |
| 1G | Control n = 6 | Unpaired Student's t test | iWAT | | 0.019621248 |
| | Vis-KO n = 6 | | gWAT | | 0.002534473 |
| | | | mWAT | | 0.000189507 |
| | | | rWAT | | 7.14636E-06 |
| 1H | Control n = 4 | Unpaired Student's t test | gWAT | Basal | 0.02847 |
| | Vis-KO n = 4 | | | | |
| 1I | Control n = 4 | Unpaired Student's t test | mWAT | Basal | 0.00457 |
| | Vis-KO n = 4 | | | | |
| 1J | Control n = 4 | Unpaired Student's t test | rWAT | Basal | 0.04901 |
| | Vis-KO n = 4 | | | | |

*Table 2 continued on next page*

*Table 2 continued*

| Figure | N (sample size) | Statistical test method | Description | | *p*-value |
|---|---|---|---|---|---|
| 1K | Control n = 4 | Unpaired Student's t test | iWAT | Basal | 0.03254 |
| | Vis-KO n = 4 | | | | |
| 2B | Control n = 4 | Unpaired Student's t test | Inguinal | *Adipoq* | 0.00251 |
| | Vis-KO n = 4 | | | | |
| 2D | Control n = 4 | Unpaired Student's t test | Gonadal | *Zfp423* | 2.1E-06 |
| | Vis-KO n = 4 | | | | |
| 2E | Control n = 4 | Unpaired Student's t test | Gonadal | *Cidea* (red bar) | 3.1573E-06 |
| | Vis-KO n = 4 | | | *Cidea* (blue bar) | 0.028888184 |
| | | | | *Dio2* | 0.039314815 |
| | | | | *Ppargc1a* | 0.00022784 |
| | | | | *Ucp1* (red bar) | 0.004431674 |
| | | | | *Ucp1* (blue bar) | 0.000356176 |
| 2F | Control n = 4 | Unpaired Student's t test | Gonadal | Basal | 6.73733E-05 |
| | Vis-KO n = 4 | | | Oligo. | 0.000488801 |
| | | | | FCCP | 0.02909696 |
| 2G | Control n = 4 | Unpaired Student's t test | Gonadal | | 0.03356 |
| | Vis-KO n = 4 | | | | |
| 2H | Control n = 4 | Unpaired Student's t test | Gonadal | *Gsta3* | 0.019517318 |
| | Vis-KO n = 4 | | | *Tcf21* | 0.014978261 |
| | | | | *Wt1* | 0.000632618 |
| | | | | *Ear2* | 0.009978141 |
| | | | | *Hoxa9* | 0.000427928 |
| | | | | *Tmem26* | 0.004299176 |
| | | | | *Zic1* | 0.034783957 |
| 3A | Control n = 5 | Unpaired Student's t test | gWAT | | 0.000759641 |
| | Vis-KO n = 4 | | rWAT | | 0.001739073 |
| 3B | Control n = 5 | | | *Cidea* | 0.000155141 |
| | Vis-KO n = 4 | | | *Dio2* | 0.009600651 |
| | | | | *Prdm16* (red bar) | 0.002457629 |
| | | | | *Prdm16* (blue bar) | 7.9394E-05 |

*Table 2 continued on next page*

*Table 2 continued*

| Figure | N (sample size) | Statistical test method | Description | | *p*-value |
|---|---|---|---|---|---|
| 3G | Control n = 5 | Unpaired Student's t test | gWAT | *Gsta3* | 0.044793954 |
| | Vis-KO n = 4 | | | *Resistin* | 0.002421899 |
| | | | | *Tmem26* | 0.011651978 |
| 5A | Control n = 5 | Unpaired Student's t test | gWAT | *Ucp1* | 0.011438323 |
| | Vis-KO n = 4 | | rWAT | *Ucp1* | 0.001188624 |
| 5B | Control n = 5 | Unpaired Student's t test | rWAT | *Cidea* | 0.000780591 |
| | Vis-KO n = 4 | | | *Dio2* | 0.002682038 |
| | | | | *Elovl3* | 0.01642267 |
| | | | | *Prdm16* (red bar) | 0.042213109 |
| | | | | *Prdm16* (blue bar) | 0.000810897 |
| | | | gWAT | *Dio2* | 0.0221 |
| | | | | *Prdm16* | 0.0127 |
| 5L | Control n = 5 | Unpaired Student's t test | t = 4 hr | | 0.014972958 |
| | Vis-KO n = 4 | | t = 5 hr | | 0.035138826 |
| 6D | Control n = 7 | Unpaired Student's t test | gWAT | *Zfp423* | 0.002716658 |
| | Vis-KO n = 6 | | mWAT | *Zfp423* | 0.042346922 |
| | | | rWAT | *Zfp423* | 0.018568608 |
| 6E | Control n = 7 | Unpaired Student's t test | iWAT | *Ucp1* | 0.002716658 |
| | Vis-KO n = 6 | | gWAT | *Ucp1* | 0.042346922 |
| | | | rWAT | *Ucp1* | 0.018568608 |
| 6F | Control n = 7 | Unpaired Student's t test | iWAT | *Cidea* | 0.044864901 |
| | Vis-KO n = 6 | | gWAT | *Cidea* | 0.011872021 |
| | | | | *Prdm16* | 0.046343663 |
| 6H | Control n = 4 | Unpaired Student's t test | gWAT | | 0.00648 |
| | Vis-KO n = 4 | | | | |
| 6I | Control n = 4 | Unpaired Student's t test | mWAT | | 0.03908 |
| | Vis-KO n = 4 | | | | |
| 6J | Control n = 4 | Unpaired Student's t test | rWAT | | 0.03982 |
| | Vis-KO n = 4 | | | | |

*Table 2 continued on next page*

*Table 2 continued*

| Figure | N (sample size) | Statistical test method | Description | | p-value |
|---|---|---|---|---|---|
| 7D | Control n = 4 | Unpaired Student's t test | O$_2$ Consumption | 24 hr | 0.020700477 |
| | Vis-KO n = 5 | | | Dark | 0.006034333 |
| 7E | Control n = 4 | Unpaired Student's t test | Heat Production | 24 hr | 0.022957861 |
| | Vis-KO n = 5 | | | Dark | 0.007378241 |
| 7F | Control n = 4 | Unpaired Student's t test | CO$_2$ Production | 24 hr | 0.041576891 |
| | Vis-KO n = 5 | | | Dark | 0.017964281 |
| 8A | Control n = 7 | Unpaired Student's t test | t = 0 min | | 0.043779825 |
| | Vis-KO n = 6 | | t = 15 min | | 0.004769743 |
| | | | t = 30 min | | 0.012903721 |
| | | | t = 60 min | | 0.014915684 |
| 8B | Control n = 7 | Unpaired Student's t test | t = 0 min | | 0.009873322 |
| | Vis-KO n = 6 | | t = 30 min | | 0.01583899 |
| 8C | Control n = 7 | Unpaired Student's t test | t = 30 min | | 0.0426 |
| | Vis-KO n = 6 | | | | |
| 8D | Control n = 6 | Unpaired Student's t test | Glucose Infusion Rate | | 0.04689 |
| | Vis-KO n = 5 | | | | |
| 8E | Control n = 6 | Unpaired Student's t test | Hepatic Glucose Prod. | Basal | 0.039627397 |
| | Vis-KO n = 5 | | | Clamped | 0.0346953 |
| 8F | Control n = 7 | Unpaired Student's t test | Liver | *Foxo1* | 0.046418578 |
| | Vis-KO n = 6 | | | *G6Pc* | 0.024438785 |
| | | | | *Pck1* | 0.047698079 |
| 8G | Control n = 7 | Unpaired Student's t test | TAG | | 0.02449 |
| | Vis-KO n = 6 | | | | |
| 8I | Control n = 4 | Unpaired Student's t test | 2-DG Uptake | rWAT | 0.04023 |
| | Vis-KO n = 5 | | | | |
| 9B | Control n = 4 | Unpaired Student's t test | Pdgfr$\beta$ cells | | 1.1E-05 |
| | iMural-KO n = 5 | | | | |

*Table 2 continued on next page*

Hepler *et al*. eLife 2017;6:e27669. DOI: 10.7554/eLife.27669

*Table 2 continued*

| Figure | N (sample size) | Statistical test method | Description | | | p-value |
|---|---|---|---|---|---|---|
| 10C | Control n = 8 | Unpaired Student's t test | gWAT | | | 0.00276 |
| | iMural-KO n = 6 | | | | | |
| 10I | Control n = 8 | Unpaired Student's t test | t = 30 min | | | 0.003328841 |
| | iMural-KO n = 6 | | t = 60 min | | | 0.048919491 |
| 10K | Control n = 8 | Unpaired Student's t test | t = 15 min | | | 0.03752 |
| | iMural-KO n = 6 | | | | | |
| 10-S1C | Control n = 7 | Unpaired Student's t test | gWAT | | | 7.8E-08 |
| | iMural-KO n = 7 | | | | | |
| 10-S1E | Control n = 8 | Unpaired Student's t test | iWAT | *Ucp1* | | 0.015960549 |
| | iMural-KO n = 6 | | gWAT | *Ucp1* | | 0.005703268 |
| | | | rWAT | *Ucp1* | | 0.011883251 |
| 10-S1F | Control n = 5 | Unpaired Student's t test | iWAT | | | 0.00019253 |
| | Vis-KO n = 5 | | gWAT | | | 0.042623514 |
| | | | mWAT | | | 0.017424031 |
| | | | rWAT | | | 0.00228672 |

## Metabolic phenotyping

Metabolic cage studies were conducted using TSE Phenomaster cages (TSE Systems, Chesterfield, Missouri) at the USTW Metabolic Phenotyping Core. Mice were acclimated in the metabolic chambers for 5 days before the start of the experiments. Food intake, movement, and $CO_2$ and $O_2$ levels were measured every 60 min for each mouse over a period of 5 days. For glucose tolerance tests, mice were fasted overnight and then administered glucose by intraperitoneal injection (1 g/kg body weight, Sigma). For insulin tolerance tests, mice were fasted for 4 hr and then administered insulin by intraperitoneal injection (0.75 U/kg body weight human insulin, Eli Lilly). At the indicated timepoints, tail blood was collected. Hyperinsulinemic euglycemic clamps were performed on conscious, unrestrained mice as previously described (*Holland et al., 2011*). Blood glucose was measured using Bayer Contour glucometers. Serum triglycerides were measured using Infinity Triglycerides Reagent (Thermo Fisher Scientific). Serum insulin was measured using Ultra Sensitive Mouse Insulin ELISA (Crystal Chem).

## Cold exposure

Mice 8 weeks of age were transferred to a 30°C chamber (thermoneutrality) for two weeks. One week before cold exposure, IPTT-300 temperature transponders (Bio Medic Data Systems) were implanted subcutaneously in the dorsal region of the mice. Body temperature was assessed using a DAS-7006/7 s reader (Bio Medic Data Systems). The mice were then transferred to the cold chamber (6°C) or maintained at thermoneutrality. For the acute cold tolerance test, food was removed from the cages upon transfer to the cold chamber and body temperature was measured at the indicated

time points. For the acclimated cold exposure test, food was removed from the cages after 4 weeks of acclimation to cold and body temperature was measured at the indicated time points.

## Mitochondrial function and respiration

Adipose tissue fragments or cultured adipocytes were assayed for oxygen consumption rate (OCR) using an XF24 Extracellular Flux Analyzer (Seahorse Bioscience, MA). Assays of mitochondrial function and respiration rates in whole adipose tissues or cells were performed as previously described (*Shao et al., 2016*). In brief, adipose tissue was cut into 5–10 mg pieces and locked into an XF24 islet-capture Microplate (Seahorse Bioscience). Adipose tissues or cultured adipocytes were equilibrated for 1 hr at 37°C in a $CO_2$-free incubator in XF Assay Medium (Modified DMEM, 0 mM Glucose; Seahorse Bioscience) (pH 7.4), supplemented with 1 mM sodium pyruvate, 1 mM L- Glutamine and 7 mM glucose. Tissues or cells were subjected to a 10 min equilibration period and three assay cycles to measure the basal rate, comprising a 3 min mix, a 2 min wait and a 3 min measure period each. For cells, compounds were then added by automatic pneumatic injection followed by assay cycles after each, comprising of 3 min mix, 2 min wait and a 3 min measure period. OCR measurements were obtained following sequential additions of oligomyin (3 μM final concentration), FCCP (9 μM) and antimycinA/rotenone (3 μM/30 nM). OCR measurements were recorded at set interval time points. All compounds and materials above were obtained from Sigma-Aldrich.

## Statistical analysis

All data were expressed as the mean ± SEM. We used GraphPad Prism 7.0 (GraphPad Software, Inc., La Jolla, CA, USA) to perform the statistical analyses. Each experiment was performed at least twice and representative data are shown. For comparisons between two independent groups, a Student's *t*-test was used and $p < 0.05$ was considered statistically significant. The sample size estimation was determined as described below. All the detailed sample sizes, statistical test methods, and *p*-values are listed in *Table 2*. Longitudinal metabolic cohorts were designed to detect a 25% improvement of glucose tolerance or a similar magnitude of insulin resistance with an assumed 15% standard deviation of the group means at a power of 80% and an alpha of 0.05. This predicted approximately six animals per test group. All animals in the cohort were subsequently used for downstream assays (gene expression, IHC, western blot) to eliminate selection bias. We estimated the approximate effect size based on independent preliminary studies. Studies designed to characterize an in vitro difference in metabolic flux were estimated to have a slightly larger effect size of 30% with assumed 15% standard deviation of group means. To detect this difference at a power of 80% and an alpha of 0.05, we predicted we would need four independent replicates per group. We estimated this effect size based on independent preliminary studies.

## Acknowledgements

The authors are grateful to members of the UTSW Touchstone Diabetes Center for useful discussions, and Drs. P Scherer, and C Kusminski for critical reading of the manuscript. The authors thank the UTSW Animal Resource Center, McDermott Sequencing Center, Metabolic Phenotyping Core, Pathology Core, Live Cell Imaging Core, and Flow Cytometry Core for excellent guidance and assistance with experiments performed here. This study was supported by the Searle Scholars Program (Chicago, IL) and NIDDK R01 DK104789 to RKG, NIDDK R00-DK094973 and JDRF Award 5-CDA-2014–185-A-N to WLH, F30 DK100095 to JYX, the American Heart Association postdoctoral fellowship, 16POST26420136, to MS, F31 DK113696-01 and NIH NIGMS T32 GM008203, to CH.

## Additional information

### Funding

| Funder | Grant reference number | Author |
| --- | --- | --- |
| National Institutes of Health | R01 DK104789 | Rana K Gupta |
| National Institutes of Health | R00-DK094973 | William L Holland |
| American Heart Association | 16POST26420136 | Mengle Shao |

| | | |
|---|---|---|
| Searle Scholars Program | | Rana K Gupta |
| Juvenile Diabetes Research Foundation | 5-CDA-2014-185-A-N | William L Holland |
| National Institutes of Health | F30 DK100095 | Jonathan Y Xia |
| National Institutes of Health | T32 GM008203 | Chelsea Hepler |
| National Institutes of Health | F31DK113696 | Chelsea Hepler |

The funders had no role in study design, data collection and interpretation, or the decision to submit the work for publication.

### Author contributions

CH, Conceptualization, Data curation, Formal analysis, Supervision, Investigation, Visualization, Methodology, Writing—original draft, Project administration, Writing—review and editing; MS, LV, Supervision, Investigation, Methodology; JYX, Data curation, Formal analysis, Investigation, Visualization, Methodology; ALG, Formal analysis, Investigation, Methodology, Writing—review and editing; MJP, TSM, Investigation, Methodology; AXS, Formal analysis, Investigation; WLH, Resources, Supervision, Investigation, Methodology; RKG, Conceptualization, Formal analysis, Supervision, Funding acquisition, Investigation, Project administration, Writing—review and editing

### Author ORCIDs

Rana K Gupta, http://orcid.org/0000-0002-9001-4531

### Ethics

Animal experimentation: This study was performed in strict accordance with the recommendations in the Guide for the Care and Use of Laboratory Animals of the National Institutes of Health. All of the animals were handled according to approved institutional animal care and use committee (IACUC) protocols (APN 2012-0072 and APN 2015-101207 ) of UTSW Medical Center.

## Additional files

### Major datasets

The following dataset was generated:

| Author(s) | Year | Dataset title | Dataset URL | Database, license, and accessibility information |
|---|---|---|---|---|
| Hepler C, Gupta RK | 2017 | Adipose tissue from $\beta$-3 agonist-treated mice | https://www.ncbi.nlm.nih.gov/geo/query/acc.cgi?acc=GSE98132 | Publicly available at the NCBI Gene Expression Omnibus (accession no: GSE98132) |

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
