## [Decision Letter]

Thank you for submitting your article "Directing Visceral White Adipocyte Precursors to a Thermogenic Adipocyte Fate Improves Insulin Sensitivity in Obesity" for consideration by *eLife*. Your article has been favorably evaluated by Fiona Watt as the Senior Editor, Peter Tontonoz as the Reviewing Editor, and two reviewers.

The reviewers have discussed the reviews with one another and the Reviewing Editor has drafted this decision to help you prepare a revised submission..

Summary:

The manuscript by Zhang et al. provides compelling evidence that visceral white fat-selective deletion of ZFP423 causes a "browning" of white fat and improves glucose homeostasis. The authors utilized two mouse models to reveal a role for *Zfp423* in visceral WAT browning. They deleted *Zfp423* specifically in visceral WAT depots by crossing *Zfp423*-floxed mice with *Wt1-Cre* mice and generated mice with inducible *Zfp423* deletion in Pdgfrβ-expressing mural cells. They also showed in cultured cell experiments that deletion of ZFP423 caused the white adipocyte browning in a cell-autonomous manner. Although the visceral WAT browning in response to ZFP423 inactivation did not affect adiposity under a high-fat diet, the KO mice exhibited improved glucose tolerance and insulin tolerance.

Given that subcutaneous WAT is considered the major site for WAT browning, the novelty of this study resides in discovery of the potential for visceral adipose tissue to undergo a thermogenic phenotype switch as a result of genetic manipulation.

The reviewers were in agreement that the work was interesting and potentially of interest to the general audience of *eLife*. The studies were judged to be technically sound and the reviewers believed that the majority of the conclusions drawn were supported by the data. At the same time, the review process identified several opportunities to strengthen the work.

Essential revisions:

1) The data in Figure 5 are convincing that glucose tolerance is improved in KO mice. It would be interesting to test the extent to which glucose uptake in the adipose tissues is enhanced, or other tissues like the skeletal muscle may be involved. It would also be insightful to test insulin sensitivity or insulin signaling in the liver and muscle.

2) In Figure 1, it is intriguing that tissue respiration among the WAT depots are similar despite the large difference in UCP1 expression (UCP1 mRNA in the inguinal WAT is much higher than those in visceral WAT depots). The authors should examine if the high OCR in the KO mice was due to uncoupling respiration (oligomycin-insensitive respiration) or not.

3) In the context of therapy, how much could visceral WAT browning contribute to systemic energy expenditure? Figure 6—figure supplement 2C showed that only about 6% of all adipocytes are multilocular in iMural-KO mice after CL treatment. With such small numbers of multilocular cells, it is impressive that the authors observed large differences in GTT and ITTs. It would be helpful if the authors could provide whole-animal metabolic chamber data to support systemic metabolic improvement of these mice.

4) The authors demonstrated that *Wt1-Cre Zfp423* KO mice lost *Zfp423* expression exclusively in visceral WAT depots. However, Figure 1 showed upregulation of thermogenic genes and OCR in not only visceral WAT but also inguinal WAT depot. Similar results on Ucp1 expression were also observed after high-fat diet feeding (Figure 5). Therefore, changes in systemic glucose metabolism and serum lipid levels shown later on in Figure 5 may not be entirely a direct result of visceral WAT browning.

5) In Figure 6 and Figure 6—figure supplement 1, the authors showed enhanced thermogenic adaptation and improved glucose tolerance in the iMural-KO mice in response to CL treatment. However, *Zfp423* deletion in this mouse model is not only in visceral WAT but also in subcutaneous WAT, because Pdgfrβ is expressed in the progenitor cells of both. Therefore, the improved metabolic phenotype cannot be exclusively attributed to loss of *Zfp423* in visceral mural cells. Have the investigators examined any effects in iWAT, for example with similar studies shown in Figure 6—figure supplement 2A-C?

---

## [Author Response]

*Essential revisions:*

*1) The data in Figure 5 are convincing that glucose tolerance is improved in KO mice. It would be interesting to test the extent to which glucose uptake in the adipose tissues is enhanced, or other tissues like the skeletal muscle may be involved. It would also be insightful to test insulin sensitivity or insulin signaling in the liver and muscle.*

This is a great suggestion, particularly as this is a unique model of visceral WAT selective browning. The reviewer asks great questions that are best answered by the use of hyperinsulinemic euglycemic clamps; this assay is the gold standard for assessing insulin sensitivity in vivo. To further explore insulin sensitivity and glucose uptake in adipose depots and other tissues, we performed the clamp assays on control and Vis-KO mice after 8 weeks of high fat diet feeding (when glucose tolerance is improved and body weight is not altered).

– The glucose infusion rate needed to maintain euglycemia (~137 mg/dl) was increased in Vis-KO mice. This demonstrates an increase in whole-body insulin sensitivity, consistent with our insulin tolerance tests.

– Importantly, hepatic glucose output was suppressed much more efficiently during the basal and clamped states in Vis-KO mice. These data indicate improved insulin sensitivity at the level of the liver.

– Tracer kinetics indicate that the rate of disposal of glucose was not significantly different between control and Vis-KO mice, suggesting minimal differences in insulin-stimulated glucose uptake by the muscle.

– However, 2-DG uptake was enhanced in the visceral rWAT depots, correlating with where the largest degree of beiging is observed.

All together these data demonstrate visceral deletion of *Zfp423* leads to enhanced insulin sensitivity and reduced hepatic glucose output. These new data are presented in the new Figure 8 of the manuscript.

*2) In Figure 1, it is intriguing that tissue respiration among the WAT depots are similar despite the large difference in UCP1 expression (UCP1 mRNA in the inguinal WAT is much higher than those in visceral WAT depots). The authors should examine if the high OCR in the KO mice was due to uncoupling respiration (oligomycin-insensitive respiration) or not.*

The reviewers raise a good point. It is not our intention to make any comparisons across WAT depots. We only wish to compare controls to KO animals, within the same depots. For this reason, we plot OCRs from different depots in different graphs. Nevertheless, to answer the reviewers’ question we think that there are technical limitations when using whole tissue fragments in the Seahorse assay that prevent us from comparing OCRs across different depots. In particular, it is difficult to compare oxygen consumption rates between different depots when considering differences in cell composition and lipid content. We are unable to assume that the drug diffusion dynamics will be the same across depots in these assays. In order to better illustrate the comparisons that we wish to convey, we have now revised the OCR graphs throughout the manuscript to represent OCRs of KO whole adipose tissue relative to those from controls (i.e. normalized to 1). This, in fact, is how others have presented these types of data in the literature (1).

More importantly, to address the question of whether elevated OCR in the KO mice is actually due to uncoupled respiration, we performed new and more comprehensive assays of adipocyte mitochondrial bioenergetics using 2D in vitro-derived cultured gonadal adipocytes.

We have now included basal, uncoupled (oligomycin), maximal (FCCP), and non-mitochondrial (rotenone/antimycin A) respiration of control and Vis-KO cultured gonadal adipocytes in Figure 2. We observed higher uncoupled respiration in Vis-KO gonadal adipocyte cultures, consistent with the notion *Zfp423*-deficient visceral adipocytes are more brown/beige-like.

*3) In the context of therapy, how much could visceral WAT browning contribute to systemic energy expenditure? Figure 6—figure supplement 2C showed that only about 6% of all adipocytes are multilocular in iMural-KO mice after CL treatment. With such small numbers of multilocular cells, it is impressive that the authors observed large differences in GTT and ITTs. It would be helpful if the authors could provide whole-animal metabolic chamber data to support systemic metabolic improvement of these mice.*

Great suggestion. Ultimately, the logistics of the pump implantation complicate metabolic chamber experiments (IACUC protocol issues). Moreover, we find that the pump implantation complicates analysis of the data normalization to lean mass. To address this question and better understand what maximal visceral browning can do, we performed new metabolic cages studies with the Vis-KO model. We assessed parameters of energy expenditure and food intake of control and Vis-KO mice after 8 weeks of HFD feeding.

– We do not detect significant differences in food intake or activity between control and Vis-KO mice.

– However, O_2_ consumption, CO_2_ production, and heat production are all elevated in the KO animals. These new data demonstrate visceral WAT deletion of *Zfp423* (and thus visceral browning) enhances energy expenditure, albeit not a strong enough degree to drive a robust difference in body weight.

All of these new data are presented in a brand new Figure 7. As eluded to by the reviewers, these data highlight the idea that beige (or beige-like) cells are likely exerting beneficial effects on glucose homeostasis independent of their impact on body weight.

*4) The authors demonstrated that Wt1-Cre Zfp423 KO mice lost Zfp423 expression exclusively in visceral WAT depots. However, Figure 1 showed upregulation of thermogenic genes and OCR in not only visceral WAT but also inguinal WAT depot. Similar results on Ucp1 expression were also observed after high-fat diet feeding (Figure 5). Therefore, changes in systemic glucose metabolism and serum lipid levels shown later on in Figure 5 may not be entirely a direct result of visceral WAT browning.*

Point well-taken! Given that the genetic perturbation is selective to visceral WAT depots, we would still argue that these visceral depots are ultimately the driver of the phenotype, and that the impact on iWAT must be secondary. As we now discuss further in the manuscript (revised Discussion section), we cannot rule out the possibility that *Zfp423* deficiency may impact the overall visceral adipose phenotype in a number of ways that influence energy metabolism. It is possible that *Zfp423*-deficient visceral adipocytes produce circulating adipokines that influence systemic metabolism and perhaps drive “secondary beiging” in other depots. The new clamp studies of Vis-KO mice (new Figure 8) shed some additional insight on this issue. First and foremost, glucose uptake is enhanced in the rWAT depots, correlating with where the visceral WAT browning is most prominent. Moreover, the data largely point to the liver as a major site of improved insulin sensitivity. Previously reported work on Prdm16 and subcutaneous WAT beiging suggest a strong connection between subcutaneous beige cells and liver health (2). For the field at large, it remains an open question as to how brown/beige adipocytes exert beneficial effects on systemic metabolism, independent of their impact on body weight. We intend to use this unique model as a tool to explore the mechanisms of how visceral WAT browning leads to improved glucose metabolism.

*5) In Figure 6 and* Figure 6—figure supplement 1*, the authors showed enhanced thermogenic adaptation and improved glucose tolerance in the iMural-KO mice in response to CL treatment. However, Zfp423 deletion in this mouse model is not only in visceral WAT but also in subcutaneous WAT, because Pdgfrβ is expressed in the progenitor cells of both. Therefore, the improved metabolic phenotype cannot be exclusively attributed to loss of Zfp423 in visceral mural cells. Have the investigators examined any effects in iWAT, for example with similar studies shown in Figure 6—figure supplement 2A-C?*

We thank the reviewers for this question and the suggestion to further the analysis of this model. We now include the suggested iWAT analysis in the manuscript (new Figure 10—figure supplement 1). The new data, long with our prior publication of this model (3), illustrate important points regarding the emergence of adipocytes from the Pdgfrβ+ lineage:

– In association with 12 weeks of HFD feeding, de novo adipocyte differentiation from the mural cell lineage is largely restricted to visceral WAT depots. Overall, ~10-15% of gonadal adipocytes within obese mice represent “new” adipocytes that emerge from Pdgfrβ+ precursors.

– In the inguinal WAT depot, only 1-2% of adipocytes represent new adipocytes emerging from Pdgfrβ+ precursors. In fact, there is very little de novo adipogenesis (from any origin) occurring in the iWAT of male HFD-fed mice (4).

– When obese mice (8 weeks HFD-fed) are treated with CL for 4 weeks, very little, if any, additional adipocytes emerge from Pdgfrβ+ precursors in either gWAT or iWAT.

– In the iMural-KO model, Dox-administration induces deletion of *Zfp423* in mural cells in all adipose tissue depots; however, this leads to the formation of *Zfp423*-deficient adipocytes predominantly in visceral WAT depots of obese mice.

– Upon CL treatment of obese animals, we now observe the presence of multilocular adipocytes in KO visceral WAT (compared to barely any in control mice).

– However, there is no difference in the number of multilocular inguinal adipocytes between CL treated control and KO animals.

As such, inactivation of *Zfp423* in the Pdgfrβ+ lineage leads to the formation of *Zfp423*-deficient visceral adipocytes that adopt a beige-like phenotype upon activation with the β3AR agonist. This selective induction of visceral browning upon β3AR treatment is associated with improved insulin sensitivity.

Nevertheless, the reviewers raise a good point: In the iMural-KO model, *Zfp423* is deleted in mural cells throughout the body. However, the dependency of the phenotype of β3AR activation does suggest that the beige cells are driving the improvements in glucose metabolism. When considered in combination with the data from the Vis-KO model, these data support the general conclusion that directing visceral adipocyte precursors to adopt a beige-like adipocyte phenotype leads to improved insulin sensitivity in obesity.

We have now revised the Discussion section of the manuscript to highlight some of the limitations to these models.

References

1) Stine RR, Shapira SN, Lim HW, Ishibashi J, Harms M, Won KJ, Seale P. EBF2 promotes the recruitment of beige adipocytes in white adipose tissue. Mol Metab. 2016;5(1):57-65. doi: 10.1016/j.molmet.2015.11.001. PubMed PMID: 26844207; PMCID: PMC4703852.

2) Cohen P, Levy JD, Zhang Y, Frontini A, Kolodin DP, Svensson KJ, Lo JC, Zeng X, Ye L, Khandekar MJ, Wu J, Gunawardana SC, Banks AS, Camporez JP, Jurczak MJ, Kajimura S, Piston DW, Mathis D, Cinti S, Shulman GI, Seale P, Spiegelman BM. Ablation of PRDM16 and beige adipose causes metabolic dysfunction and a subcutaneous to visceral fat switch. Cell. 2014;156(1-2):304-16. doi: 10.1016/j.cell.2013.12.021. PubMed PMID: 24439384; PMCID: PMC3922400.

3) Vishvanath L, MacPherson KA, Hepler C, Wang QA, Shao M, Spurgin SB, Wang MY, Kusminski CM, Morley TS, Gupta RK. Pdgfrbeta+ Mural Preadipocytes Contribute to Adipocyte Hyperplasia Induced by High-Fat-Diet Feeding and Prolonged Cold Exposure in Adult Mice. Cell Metab. 2016;23(2):350-9. doi: 10.1016/j.cmet.2015.10.018. PubMed PMID: 26626462; PMCID: PMC4749445.

4) Wang QA, Tao C, Gupta RK, Scherer PE. Tracking adipogenesis during white adipose tissue development, expansion and regeneration. Nat Med. 2013;19(10):1338-44. doi: 10.1038/nm.3324. PubMed PMID: 23995282; PMCID: PMC4075943.